# Assessing the energy transition in China towards carbon neutrality with a probabilistic framework

Shu Zhang ⓘ [1] & Wenying Chen ⓘ [1✉]

A profound transformation of China's energy system is required to achieve carbon neutrality. Here, we couple Monte Carlo analysis with a bottom-up energy-environment-economy model to generate 3,000 cases with different carbon peak times, technological evolution pathways and cumulative carbon budgets. The results show that if emissions peak in 2025, the carbon neutrality goal calls for a 45–62% electrification rate, 47–78% renewable energy in primary energy supply, 5.2–7.9 TW of solar and wind power, 1.5–2.7 PWh of energy storage usage and 64–1,649 $MtCO_2$ of negative emissions, and synergistically reducing approximately 80% of local air pollutants compared to the present level in 2050. The emission peak time and cumulative carbon budget have significant impacts on the decarbonization pathways, technology choices, and transition costs. Early peaking reduces welfare losses and prevents overreliance on carbon removal technologies. Technology breakthroughs, production and consumption pattern changes, and policy enhancement are urgently required to achieve carbon neutrality.

[1]Institute of Energy, Environment and Economy, Tsinghua University, 100084 Beijing, PR China.  ✉email: chenwy@tsinghua.edu.cn

The Paris Agreement requires that countries progressively update their nationally determined contributions (NDCs) and reach a balance between sources and sinks of greenhouse gases (GHGs), implying the need to achieve net-zero emissions at some point in time[1]. China is currently the world's largest energy consumer and $CO_2$ emitter, accounting for 23% and 29% of global energy consumption and $CO_2$ emissions in 2019, respectively[2]. A reduction in China's $CO_2$ emissions would constitute an incomparable contribution to climate change mitigation. China updated its climate goal in September 2020 to strive to achieve peak $CO_2$ emissions by 2030 and work towards carbon neutrality by 2060. However, considerable uncertainties in the national cumulative carbon budget and technology evolution have produced broad variability in technology choice, transition costs, and the feasibility of different decarbonization pathways towards carbon neutrality, which have different consequences for national development and global temperature control. Decision-making under high uncertainty presents a difficult challenge to policymakers[3–6]. A roadmap that comprehensively considers these uncertainties is critically needed to guide this transition[7].

Most current energy-environment-economy models adopt a scenario analysis approach to make a priori assumptions about future trends. The variety of projections for future development in different studies frequently produce widely divergent and puzzling results. Although some consensus can be obtained through model comparisons, it is challenging to capture the impacts of policy and technology development uncertainty in a probabilistic manner[8]. Additionally, it is difficult to obtain stable and reliable results using traditional sensitivity analysis in the context of multidimensional uncertainty[9]. Monte Carlo analysis (MCA) is a classical approach for climate change assessment under uncertainty[10,11] and is widely used with energy system models[12–16].

In this work, we combine MCA with the China-TIMES model, which has been extensively used in energy planning and climate change mitigation studies[17–29], to develop an improved version named China-TIMES-MCA (Fig. 1) and quantify the effects of technological development and cumulative carbon budget uncertainty. With the help of the advanced data analysis tools, the model delineates numerous possible sectoral decarbonization pathways. The results reflect the unique impacts of China's cumulative carbon budget and emission peak time on technology choice and transition costs. Over 160 GW per year of variable renewable energy on average must be deployed by 2050, irrespective of the cumulative carbon budget. Hydrogen and bioenergy with carbon capture and storage (BECCS) should be deployed on a large scale after 2040. The installed capacity and cost changes associated with these technologies significantly impact the transition cost. Deep decarbonization brings about shifts in production and consumption patterns, and the significant synergistic effects of $CO_2$ mitigation on air pollutant reductions are observed. Based on the model results, we urge enhanced policy, decisive actions, further technological innovation, and cross-sectoral collaboration in the future.

## Results

**$CO_2$ emission reduction pathways**. $CO_2$ emissions are the bulk of China's GHG emissions and the focus of this study. Three mitigation scenarios (PEAK20, PEAK25, and PEAK30) with different emission peak times are designed with 1000 cases each to highlight the urgency of an early peak. Series of least-cost technology and fuel mixes that meet the energy service demand are obtained. Figure 2 reports the variations in $CO_2$ emission reduction pathways resulting from the differences in the peak time and cumulative emissions. If the existing NDC target is tracked, emissions will peak at 10.4 $GtCO_2$ in 2030 and then decline steadily to 7.3 $GtCO_2$ in 2050, with China's cumulative emissions totaling 381.1 $GtCO_2$ from 2010 to 2050. Deep decarbonization is required to reach emission pathways consistent with the carbon neutrality goal. If no further climate action is taken by 2030 (the PEAK30 scenario), then a plunge in emissions between 2030 and 2050 is inevitable, accompanied by −0.9 to 1.6 $GtCO_2$ of emissions in 2050. The cumulative emissions of the feasible cases (675 out of 1000) are all above 259 $GtCO_2$, suggesting that it will be difficult to reach the 1.5-degree goal without substantial negative emission technologies (NETs) in the second half of this century, which is in line with previous multimodel comparison results[30,31]. Peaking earlier and lower can alleviate the pressure associated with the subsequent transition, but also frames the great challenge of near-term mitigation actions. For the pathway with an emission peak at 10.3 $GtCO_2$ in 2025, we estimate an emission reduction of 18% (over a 13–23% range) over 2025–2030 and the majority of the cases will achieve near-zero emissions (−0.9 to 2.7 $GtCO_2$) in 2050. Peaking in 2020 corresponds to the most intense near-term pressure with a 20–33% reduction from NDC in 2030, but almost no net-zero emissions are required in 2050 to support a cumulative emission of 240 $GtCO_2$ in the PEAK20 scenario. A paradox can be observed in the trade-off between early mitigation actions (greater mitigation contribution) and long-term dependence on NETs (earlier achievement of carbon neutrality).

These aggregate pathways, however, neglect the dynamics of the emission reduction processes among sectors. Taking the emission peak time as an example, the power sector, the current largest emitter, has the greatest near-term mitigation potential, and the timing of its peak determines the timing of the total $CO_2$ emissions peak. Except fossil fuel and industrial (FFI) emissions from the industry sector which might reach a peak at 4.6 $GtCO_2$ by 2020 (of which 4.0 $GtCO_2$ are energy-related emissions), the emissions from the building sector and transport sector are expected to peak in 2025–2030 at 0.8–1.0 $GtCO_2$ and in 2030–2035 at 1.3–1.4 $GtCO_2$, respectively. After peaking, we observe steep emission reductions from the power sector until 2035 with little variation across cases. Although the emission peak time of the power sector varies considerably in different scenarios, one highly consistent finding is that power sector emissions typically become negative in 2040–2045, with emissions of −1.3 to 0.1, −1.6 to 0.2, −1.6 to −0.5 $GtCO_2$ in 2050 for the PEAK20, PEAK25, and PEAK30 scenarios. This finding highlights the need to decarbonize the power sector in a timely manner. The later we act, the less time we have left to transition the power system. Turning to the demand sectors, the motivation to reduce emissions is mainly from cumulative emission limitations, and a remarkable spectrum of emission reduction pathways exists. Through the development of alternative materials, popularization of carbon capture and storage (CCS) technology and declining demand for energy-intensive products, as seen from the result for 2050, energy-related emissions from the industry sector will drop to 0.4–1.7 (PEAK20), 0.3–1.5 (PEAK25), and 0.3–1.1 $GtCO_2$ (PEAK30), and industrial process emissions will decrease by 72–91% compared to the emissions in 2020 to as low as 0.1 $GtCO_2$. The building and transport sectors have quite different statuses. Sluggish growth in building floor area and a high electrification rate in the building sector, while the continuous growth of the transport demand and less than 4% electrification rate in the transport sector, has resulted in very divergent development trends. The later the action is taken, the greater the range of 2050 emissions in the building sector (0.1–0.6 $GtCO_2$ for PEAK20 and 0.0–0.3 $GtCO_2$ for PEAK30), while the opposite is true for the transport sector (0.6–0.9 $GtCO_2$ for PEAK20 and 0.2–0.9 $GtCO_2$ for PEAK30). It is also worth noting that the share

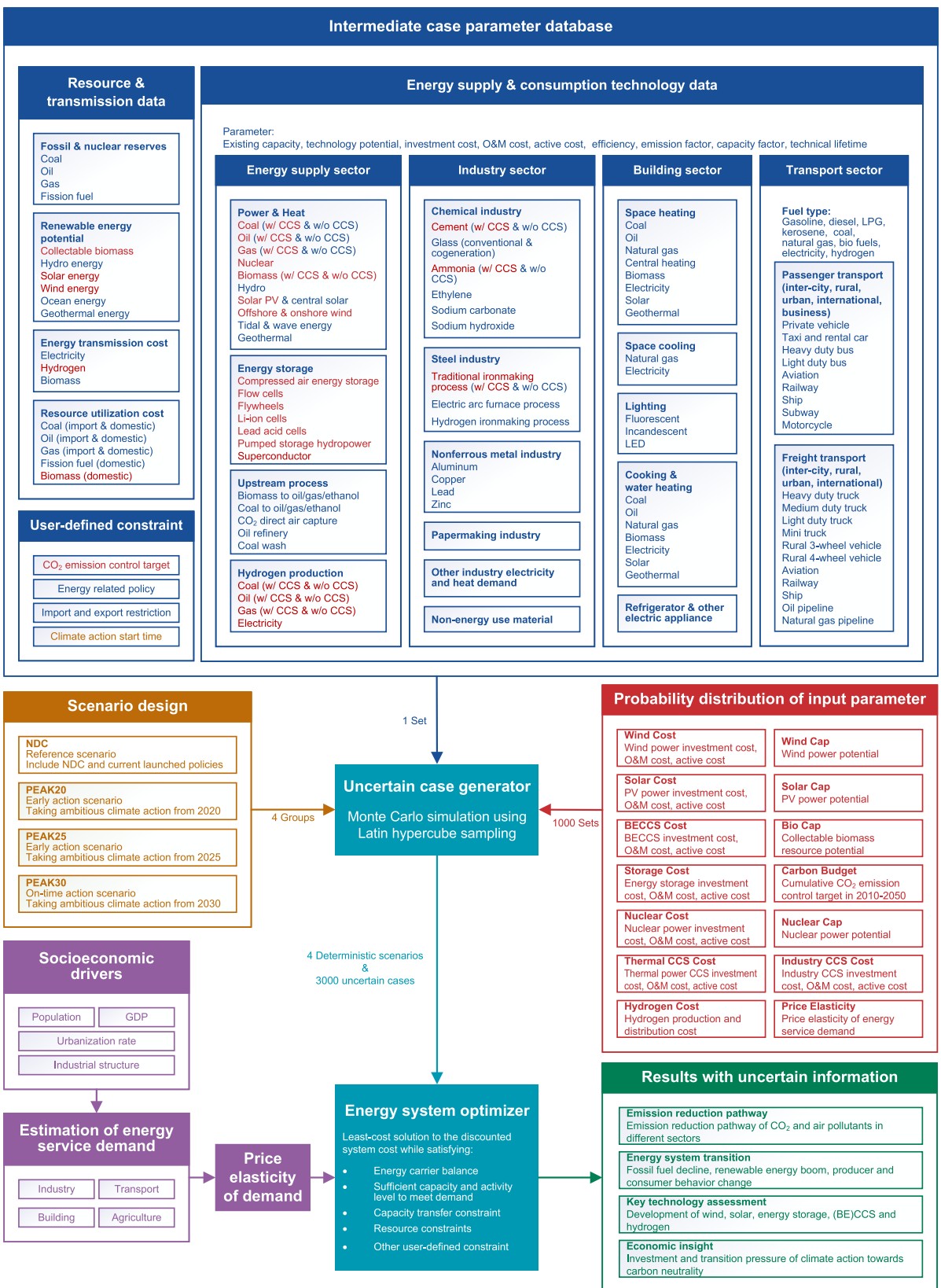

**Fig. 1 China-TIMES-MCA structure.** This figure shows the necessary information about the China-TIMES-MCA model. The cyan part is the uncertain case generator and energy system optimizer. The blue part shows the parameters obtained from the statistical data and literature for the intermediate case. The brown part illustrates the scenario design of the study. The purple part is the end-use demand considering price elasticity. The red part is the fourteen input parameters with the corresponding probability distributions. The green part is the model result, which corresponds to the main findings in this paper. In this figure, w/ CCS means that this technology is equipped with carbon capture and storage, while w/o CCS means that it is not equipped.

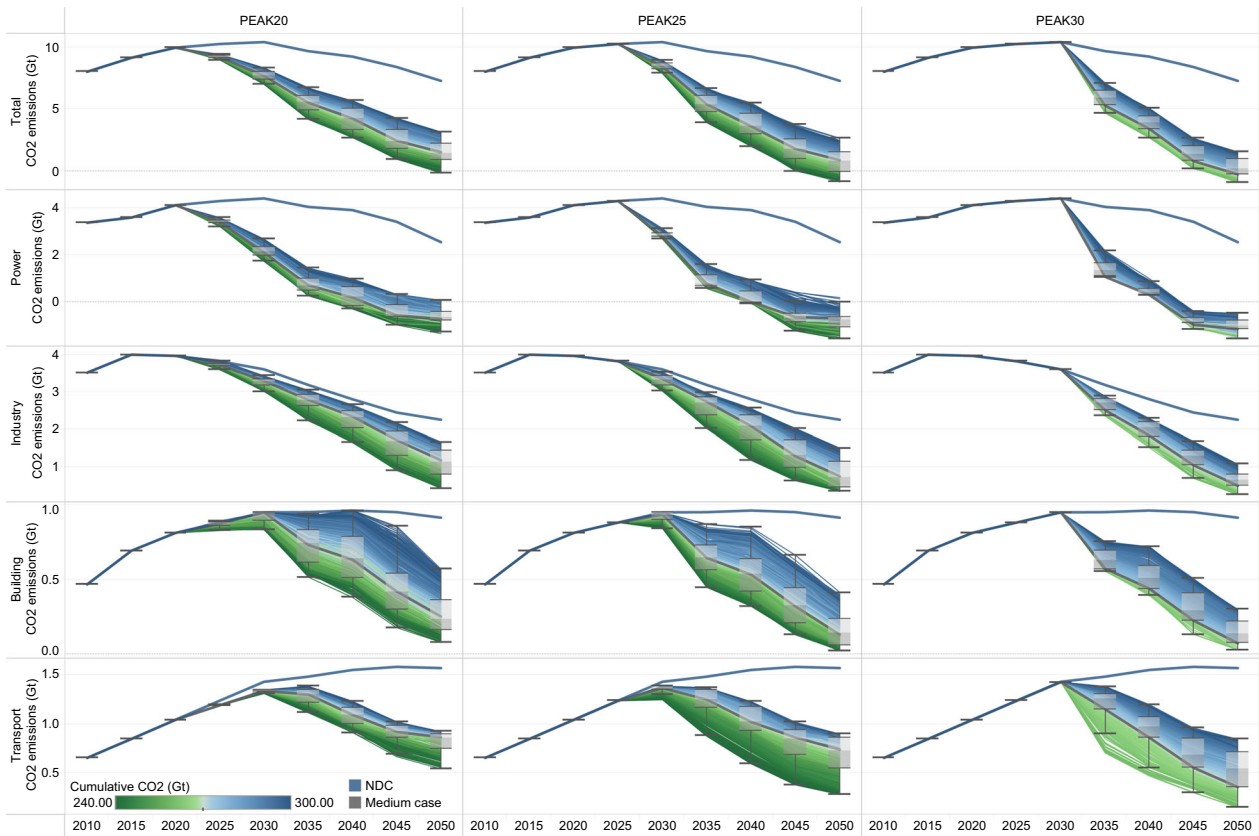

**Fig. 2 Sectoral energy-related CO₂ emission pathways under broad cumulative carbon budget range under different scenarios (unit: GtCO₂).** The box plot shows the first quantile, intermediate range (IQR), and third quantile of all the results, where the data range within 1.5 times the IQR is denoted with whiskers. The thick blue line represents the emission pathway of the NDC scenario, and the thick gray line represents the emission pathway for the intermediate case in each scenario. The divergent color from blue to green reflects the increasing stringency of the cumulative carbon budget. The cumulative CO₂ parameter corresponds to the absolute value of China's cumulative carbon budget for 2010–2050.

of transport sector emissions amongst all demand sectors will rise from 18% in 2020 to 46% (over a 32–59% range) in 2050 under the PEAK25 scenario, bringing it close to or above the industry sector's share. This finding suggests that the building sector needs to act faster than the transport sector in the near term, but transport sector emission reductions require full attention in the long term.

For China's carbon neutrality goal of net-zero GHG emissions from all economic sectors by 2060, a significant reduction in non-$CO_2$ GHGs is imperative. Given the extreme uncertainty of land use, land-use change and forestry (LULUCF) emissions, China's $CH_4$, $N_2O$ and F-gas emissions, excluding LULUCF, were 2.4 $GtCO_2e$ in 2020, and at least a 50% reduction is needed by 2050 to meet stringent climate targets[32]. Since non-$CO_2$ GHG emissions are considered difficult to completely remove based on current perceptions, more ambitious $CO_2$ reductions are required to achieve a 1.5-degree and China's carbon neutrality target.

**Energy supply decarbonization.** Behind the emission reductions are profound changes in the energy system, especially in the energy supply sector. The consistent results of all scenarios reflect that the basis of China's energy system transformation is eliminating fossil fuel consumption and substantially increasing the use of new and renewable energy (Fig. 3). Considering the PEAK25 scenario as an example, both primary energy and coal consumption peak in 2025 and then fall rapidly. The share of renewable energy in primary energy supply rises from 10% in

2020 to nearly 28% (range 22–34%) in 2035 and then further to 59% (range 47–78%) in 2050, with photovoltaic (PV) and wind power playing unique roles. The elimination of fossil fuels occurs faster in PEAK20 than in PEAK25 and slower in PEAK30 than in PEAK25.

Beyond wind and solar power, nuclear power and CCS technologies are considered promising solutions to achieve a zero-carbon energy system in the future. To better understand the effects of the large uncertainties associated with these technologies themselves or externally on their development, we designed elaborate Monte Carlo simulations. Figure 4 shows the technology development trends and the uncertain influential factors. Figure 5 shows the probability distributions. In 2030, the installed capacity of wind and solar reaches 1.2 TW under the NDC scenario and ~2 TW under the PEAK20 and PEAK25 scenarios, suggesting that renewable energy development is not on track for achieving carbon neutrality based on the current NDC target. In 2050, the explosive growth of solar (4.3 TW, for a 3.0–5.5 TW range) and wind (2.4 TW, for a 1.9–3.1 TW range) will lead to ~90% of the installed capacity and electricity output of the power system from renewable sources. The solar and wind capacities are affected by the national cumulative carbon budget, investment cost, and economic potential of resources (Fig. 4a–d). Since PV and wind power are in high demand in all mitigation scenarios, they remain the largest investment component in the energy supply sector, despite their rapidly declining costs. From 2020 to 2050, for the 25th–75th percentile cases in the PEAK25 scenario, 1.5–1.8 and 2.7–3.1 trillion US dollars are required for investment in PV and wind power. If emissions peak in 2020, investments in

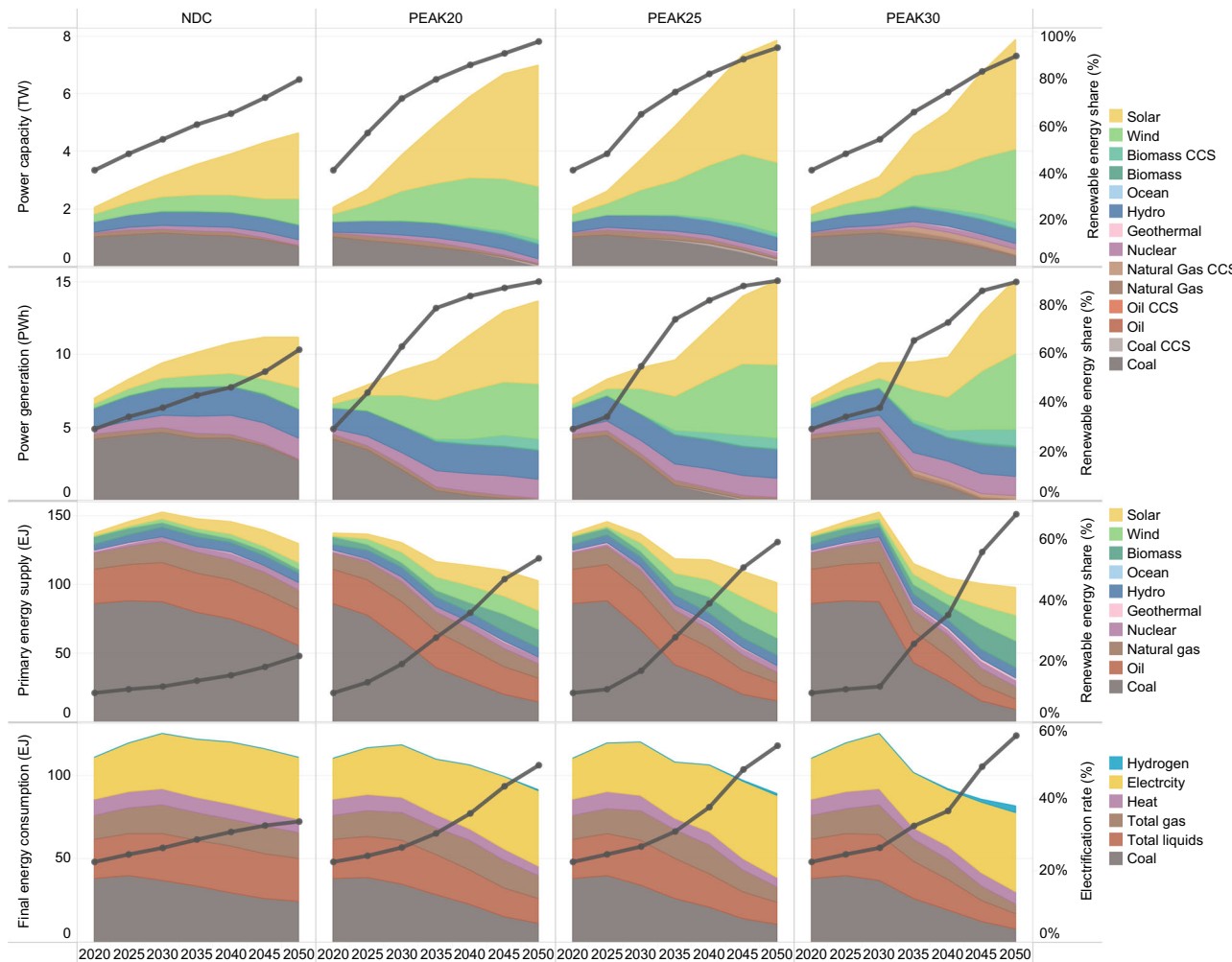

**Fig. 3 Graphic of the power installed capacity (unit: TW), the power generation (unit: PWh), the primary energy mix (unit: EJ), and the final energy mix (unit: EJ) for the intermediate case in each scenario.** For the power installed capacity, the stacked area chart shows different power plant types, and the black line represents the proportion of the renewable energy contribution to the total capacity. For the power generation, the stacked area chart shows the annual power generation of different types of power generation technologies. The black line represents the proportion of renewable energy power generation to the total power generation. In the legend, the fuel names ending in CCS indicate power plants equipped with CCS, and otherwise indicate power plants without CCS. For the primary energy mix, the stacked area chart represents different types of primary energy supply, and the black line represents the renewable energy share. The calorific value calculation method is applied for primary energy statistics. For the final energy mix, the stacked area chart shows the end-use of different energy types, and the black line represents the electrification rate.

PV and wind power will need to increase by 19–25% over the next 10 years compared to those for PEAK25, with the interquartile range spreading between 1.3 and 1.4 trillion US dollars. Additionally, we note that the elasticity of the PV and wind capacities to all uncertain input parameters are lower for the PEAK30 scenario than for the other two scenarios, suggesting that the early peak gains time for energy supply decarbonization, thereby relieving pressure on extensive wind and solar deployment beyond 2030.

The high penetration of variable renewable energy creates an enormous demand for energy storage. Figure 4e and f show that energy storage usage rapidly increases until 2045, followed by a slowdown in growth to ~15% of total generation in 2050, requiring 183–220 billion US dollars in investment during 2020–2050. The complementary relationship between energy storage technologies and PV power is shown in Supplementary Fig. 1, and an interesting shift in the role played by wind power is observed in our study. Wind power and PV power with energy storage have a clear complementary relationship, jointly replacing fossil fuel before 2040. After 2040, wind power competes with PV

power with energy storage for the night-time load supply (Supplementary Figs. 1 and 2). We also note that hydrogen production and electric vehicle (EV) charging might affect the load characteristics, such that load management can significantly reduce the demand for energy storage.

Rapid development in renewable energy is projected to occur with the phase-out of coal. If emissions do not peak before 2030, the coal share in primary energy supply needs to sharply decline from 57% in 2030 to 10% in 2050. Significant heterogeneity between power generation and the installed capacity reflects the rapidly declining capacity factor of coal-fired power, from 0.4 in 2025, to 0.3 in 2030 and 0.1 in 2035, further exacerbating the urgent issue of thermal power retirement and transition. The cautionary tale is that coal-fired power is rebounding at a time when coal-fired power needs to be controlled for development. Compared to pre-pandemic levels, coal-fired power generation growth in China outpaced wind and solar power in the first half of 2021, and China's energy supply sector did not achieve "green recovery", as hoped[33]. With 292 GW of new coal-fired power plants currently announced, permitted, shelved, and under

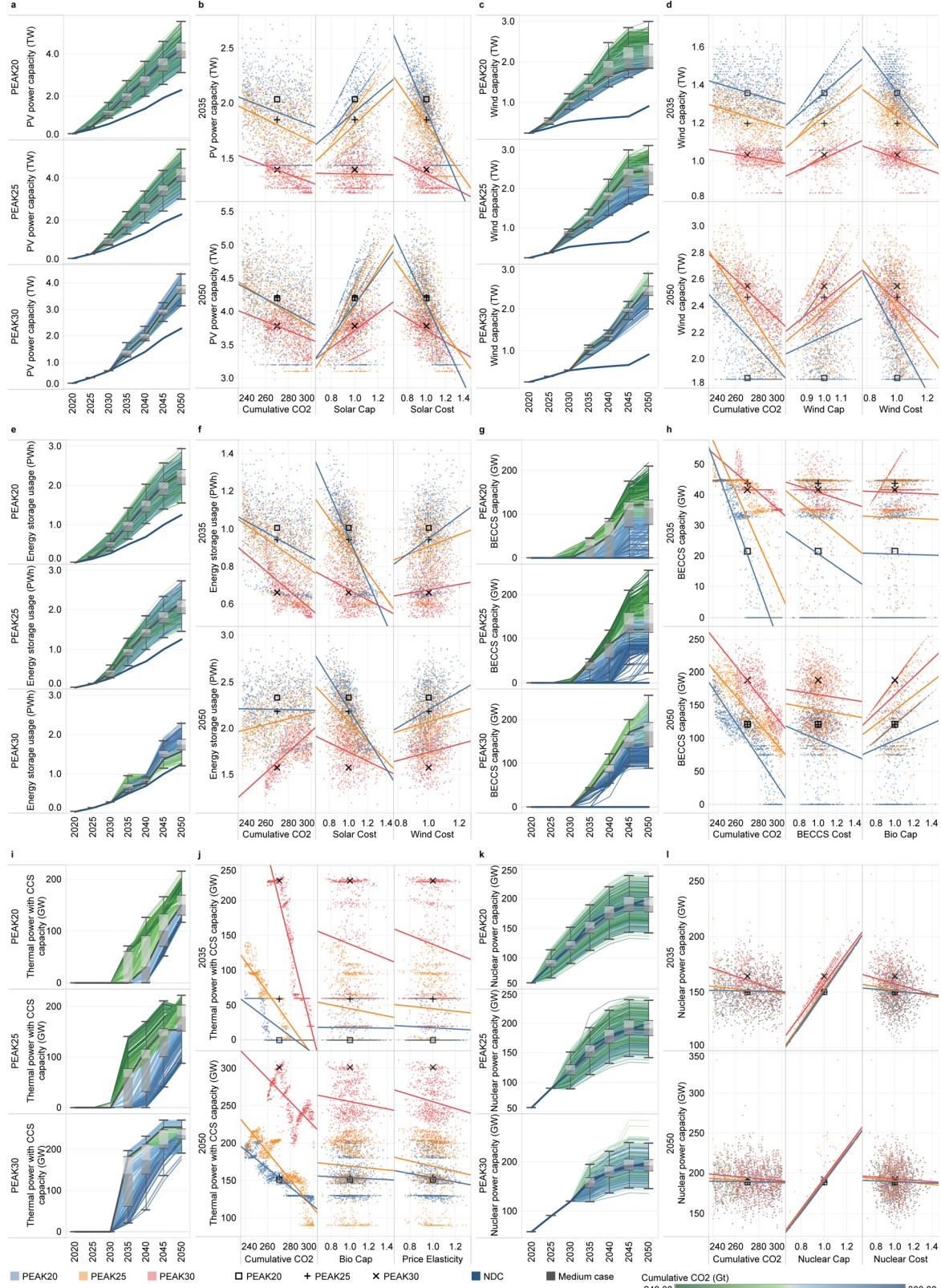

construction in China[34], stakeholders need to reassess the long-term risks involved.

Carbon lock-in and asset impairment issues for existing coal-fired power units can be effectively mitigated through flexibility retrofits, carbon capture retrofits, or upgrading to BECCS units[35]. BECCS retrofits can be made to existing thermal power plants,

and valuable negative emissions can be generated. Due to the high investment and operation cost, BECCS deployment depends largely on the national cumulative carbon budget (Fig. 4g, h). For mitigation scenarios, the application of BECCS begins in 2035 and is promoted on a large scale after 2040. Weak near- and mid-term action will cause BECCS reliance, as reflected by a

**Fig. 4 Uncertainty analysis of the development of promising technologies. a** The PV power installed capacity (unit: TW). **b** The three most critical uncertain parameters for PV power. **c** The wind power installed capacity (unit: TW). **d** The three most critical uncertain parameters for wind power. **e** The annual energy storage technology usage (unit: PWh). **f** The three most critical uncertain parameters for annual energy storage technology usage. **g** The BECCS power installed capacity (unit: GW). **h** The three most critical uncertain parameters for BECCS power. **i** The thermal power with CCS installed capacity (unit: GW). **j** The three most critical uncertain parameters for thermal power with CCS. **k** The nuclear power installed capacity (unit: GW). **l** The three most critical uncertain parameters for nuclear power. The box plot shows the first quantile, intermediate range (IQR), and third quantile of all the results, where the data range within 1.5 times the IQR is denoted with whiskers. The thick blue line represents the pathway of the NDC scenario, and the thick gray line represents the pathway for the intermediate case in each scenario. The divergent color from blue to green reflects the increasing stringency of the cumulative carbon budget. The cumulative $CO_2$ parameter corresponds to the absolute value of China's cumulative carbon budget for 2010–2050. The scatter plots depict the relationship between the development of each technology and uncertain input parameters, as fitted by a linear function. Three parameters with significant impacts among the fourteen uncertain input parameters are shown for each technology. The remaining results are shown in Supplementary Figs. 5–10. The variables (except for the cumulative carbon budget) represent multiples of the intermediate cases for each scenario. The intermediate cases of PEAK20, PEAK25, and PEAK30 are denoted with square, plus, and multiplication signs, respectively. PV photovoltaic, BECCS bioenergy with carbon capture and storage, CCS carbon capture and storage.

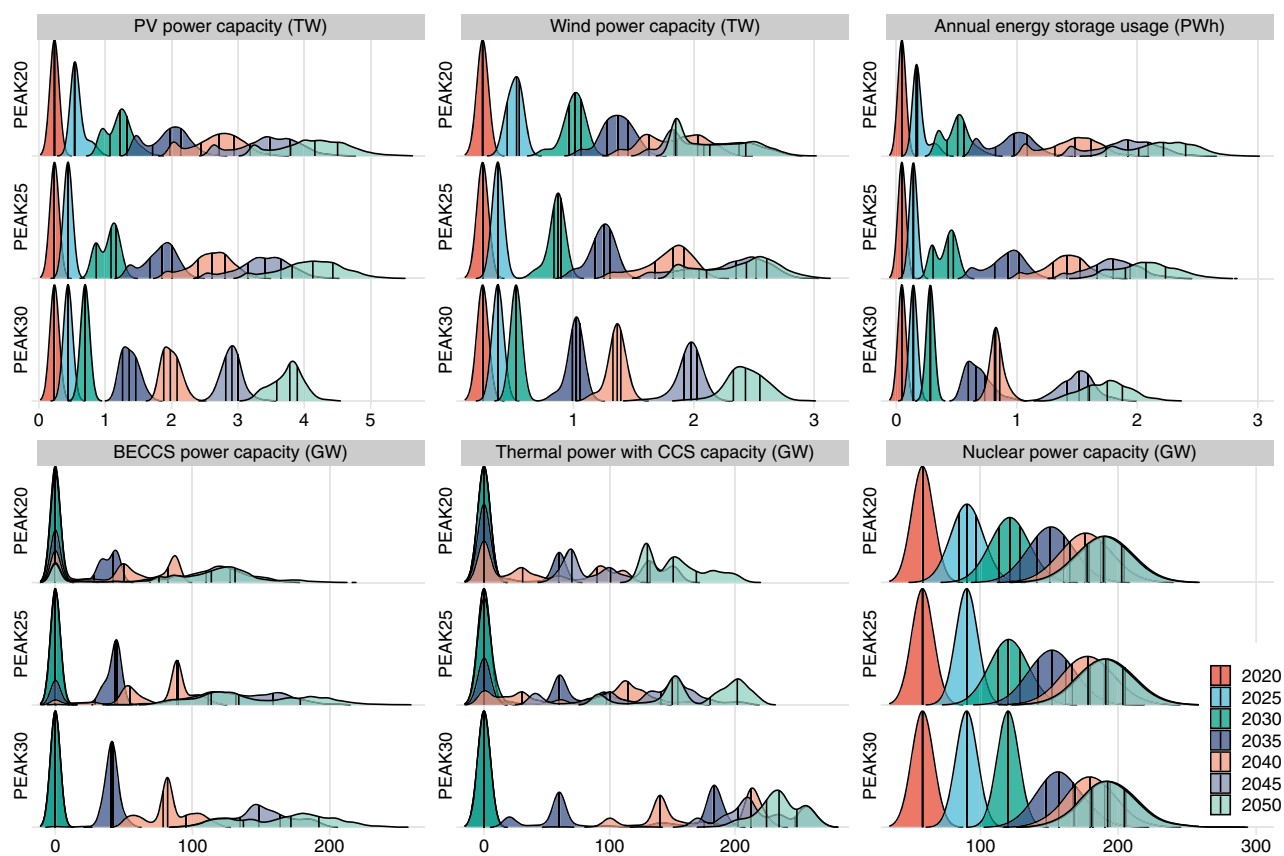

**Fig. 5 Probability density distribution of promising technologies in different periods.** Figure shows the kernel density estimation results for the PV power installed capacity (unit: TW), wind power installed capacity (unit: TW), annual energy storage technology usage (unit: PWh), BECCS power installed capacity (unit: GW), thermal power with CCS installed capacity (unit: GW) and nuclear power installed capacity (unit: GW). The black line in the graph represents the first quantile, intermediate range (IQR), and third quantile for all the cases. PV photovoltaic, BECCS bioenergy with carbon capture and storage, CCS carbon capture and storage.

comparatively higher capacity and an earlier start of large-scale deployment. Our estimate suggests that the full utilization of biomass resources suitable for power generation can generate the maximum negative emissions of 1.6 GtCO$_2$ (Supplementary Figs. 3 and 4). Increasing upfront action would reduce the biomass demand to half of the biomass potential. Changes in the biomass potential and BECCS investment cost can significantly influence the prospects for BECCS technology development. Despite being a stop-gap measure, thermal power with CCS also displays growth opportunities beyond 2035, providing power system flexibility and some heating needs. The development of thermal power with CCS units corresponds to smooth gradient of

increasing capacity while decreasing the cumulative carbon budget (Fig. 4i, j). For the 25$^{th}$–75$^{th}$ percentile cases, thermal power with CCS capacity reaches 151 GW (over a 130–169 GW range) for PEAK20 and 154 GW (over a 150–202 GW range) for PEAK25 in 2050, most of which is retrofitted from existing thermal power. For the PEAK30 scenario, 225–249 GW of thermal power with CCS is needed in 2050, and most of it will be gas-fired power with CCS to further reduce emissions.

Nuclear power, with a high capacity factor and excellent supply reliability, will grow rapidly in the next 20 years as the ballast of the power system. There are 31 coastal sites and 46 inland sites available for nuclear power construction; in other words,

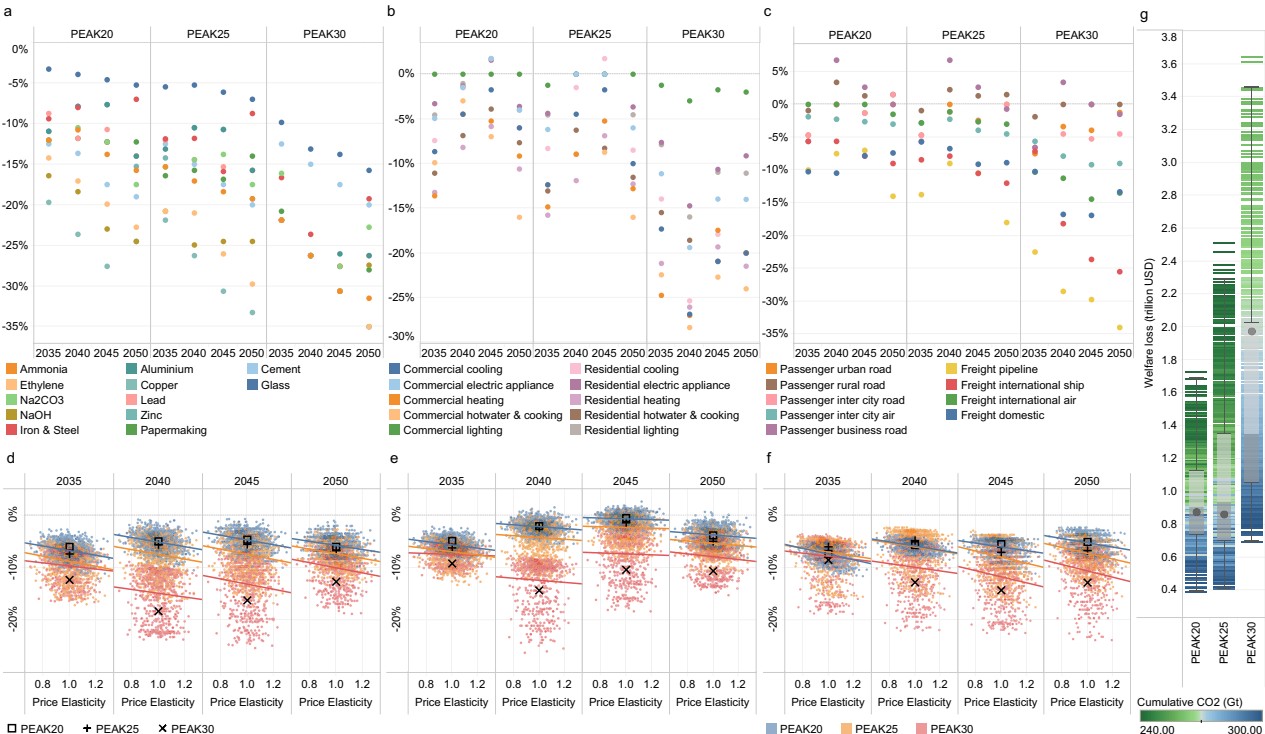

**Fig. 6 Graphic of the impact of mitigation on producer and consumer behaviors for the intermediate case in each scenario. a–c** show the declines in the rate of energy service demand for the industry, building, and transport sectors due to price elasticity relative to the NDC scenario, respectively. **d–f** represent the relationship between the energy service demand for the industry, building, and transport sectors and uncertain price elasticity, respectively. The intermediate cases of PEAK20, PEAK25, and PEAK30 are denoted with square, plus, and multiplication signs, respectively. **g** summarizes the total discounted welfare loss relative to the value in the NDC scenario for 2020–2050 (unit: trillion US dollar). The black points in panel **g** denote the welfare loss for the intermediate case in each scenario. The box plot shows the first quantile, intermediate range (IQR), and third quantile of all the results, where the data range within 1.5 times the IQR is denoted with whiskers. The divergent color from blue to green reflects the increasing stringency of the cumulative carbon budget. The cumulative $CO_2$ parameter corresponds to the absolute value of China's cumulative carbon budget for 2010–2050. $Na_2CO_3$ sodium carbonate, NaOH sodium hydroxide.

~200 GW and ~250 GW of coastal and inland nuclear power can be built[36]. Limited site location is a major constraint of nuclear power development (Fig. 4k, l). For the intermediate cases in all scenarios, the installed capacity of coastal nuclear power reaches 176 GW in 2040, which is 3.5 times higher than that in 2020. By 2050, the installed capacity rises slightly to 190 GW, and the sites available for nuclear power construction along the coast are largely built out. Variation among cases is the result of the choice of nuclear unit capacity at each site. The results also show that inland nuclear power has the potential to reach a modest scale of construction (less than 150 GW) if extremely stringent cumulative emission targets are established, renewable energy development is hindered, and nuclear power costs rapidly decline in the PEAK30 scenario. Moreover, in China's most recent 14th Five-Year Plan, inland nuclear power development is not discussed. Thus, the future advancement of inland nuclear power may be limited.

Through an analysis of a large number of cases, the following robust findings were obtained. First, renewable energy should and will grow fastest among the energy supply technologies in mitigation scenarios compared to the NDC scenario (without further policies). Second, after 2030, coal-fired power without CCS will rapidly diminish, and BECCS will gradually gain popularity after 2035 and become crucial by the middle of this century. Third, thermal power with CCS and energy storage technologies that can provide power system stability and reliability in the future will receive increased development attention. Fourth, with the exception of nuclear power, which

has limited potential due to siting constraints, all other technological developments are greatly influenced by the commitment to climate action.

**Energy demand decarbonization.** While the energy transition is more urgent in the energy supply sector, the transitions in the demand sectors cannot be downplayed. Production mode and consumption pattern shifts, technological efficiency improvements and fuel substitution are the three pillars of decarbonization in the demand sectors.

As a result of the first two factors, final energy consumption peaks by 2030 in all scenarios (Fig. 3). The production mode strongly influences the industry sector's energy service demand, and the consumption pattern or consumer behavior heavily impacts the energy service demand in the building and transport sectors. The decline in energy service demand is indicative of shifts in production modes as well as changes in consumption patterns (Fig. 6). For example, for the 25th–75th percentile cases in the PEAK30 scenario compared to those in the NDC scenario, due to industrial restructuring, energy-intensive industry production decreases rapidly, with cement and glass falling by 15–19%, steel by 11–19%, chemicals by 23–30%, and non-ferrous metals by 20–28% in 2050. The useful energy demand for heating and cooling in 2050 is estimated to decrease by 13–21% due to energy conservation technology improvements, building envelope upgrades, and modest per-capita floor space growth. The turnover of light-duty vehicles in 2050 could decrease by up to 9% in reference to the trend in the NDC scenario. Uncertainty in

price elasticity has a greater impact on industry and transport sectors but a smaller impact on the building sector probably due to the comparatively longer life span of equipment in the building sector. The closer we get to carbon neutrality, the greater the uncertainty of demand changes. This result stems from stringent climate policies increasing the cost of energy services in high-emitting sectors, thus reducing the demand due to the price rise; additionally, as society has evolved, there have become more alternative options to meet energy service demands, thus significantly increasing price elasticity. More elastic demand, while reducing energy consumption and the costs of achieving deep emission reductions to a greater extent, entails a faster rise in welfare losses (Supplementary Fig. 11). The transition caused by price changes is far from comparable to that caused by social changes, and a significant social cost could be observed. Profound changes in lifestyles and consumption concepts due to remote work, information connectivity, and the sharing economy will aid in decarbonizing energy consumption.

Electricity and hydrogen are commonly recognized as alternative energy sources in energy demand sectors. Electricity substitution is the major way to decarbonize all demand sectors. Under the PEAK25 scenario, the electrification rate reaches 55% (range 45–62%) in 2050, where the highest contributions are from electric vehicles in roadway passenger transport (from 2% today to 93% in 2050), electric space heating (from 17% today to 85% in 2050), and electric furnace steel production (from 11% today to 60% in 2050). The large gap between the present and the future suggests that electrification should be accelerated and enhanced in the future, as noted in the literature[37]. Although many studies have suggested that hydrogen is an indispensable option for decarbonization in a short period of time, different studies have not yet reached a consistent conclusion on the quantities of hydrogen energy that can be used in the demand sectors, mainly due to the high production costs and high infrastructure investments required for hydrogen transportation and distribution. Figure 7 reports our uncertainty analysis for hydrogen energy. First, we find that climate goals are the most important factors that influence the expansion of hydrogen energy. Specifically, the use of hydrogen energy exhibits a significant exponential relationship with $CO_2$ emissions, and a decrease in the cost of hydrogen energy can increase the demand for hydrogen energy to some extent. In 2050, hydrogen consumption (excluding industrial feedstock) ranges from 0.4–2.3 EJ (PEAK20), 0.4–5.4 EJ (PEAK25) and 0.4–6.0 EJ (PEAK30). The later the emission peak is reached, the lower the emissions in 2050, and the higher the demand for hydrogen energy. Second, unless the cost of electrolytic hydrogen production falls more than expected (70% lower than today by 2050), hydrogen energy will gain momentum primarily in areas lacking low-cost abatement options. For example, more than 80% of hydrogen energy is used in the transport sector, with approximately 10% used in the industry sector, and the rest used in the power and building sectors. For the 25th–75th percentile cases in the PEAK25 scenario, hydrogen fuel cell vehicles account for 33% (over a 10–64% range) of roadway freight transport (Supplementary Fig. 13), and hydrogen direct reduced iron (DRI) technology accounts for 8% (over a 4–37% range) of iron production (Supplementary Fig. 14). For the PEAK25 scenario, although hydrogen-powered aircraft may emerge after 2035, even under the strictest carbon budget constraints, the corresponding share will not exceed 41% in 2050 (Supplementary Fig. 15). Third, for the source of hydrogen, the model yields very consistent results. In China, electrolytic hydrogen production from renewable energy sources (green hydrogen) will become mainstream.

For the steel and cement industries, carbon capture and storage (CCS) technology is a low-cost option for decarbonization. In

2050, the CCS penetration rate in ironmaking reaches 67% (over a 2–80% range) and 77% (over a 25–86% range) of cement production applies CCS technology in the PEAK25 scenario (Supplementary Figs. 14 and 16). Since CCS technologies require the extensive construction of pipelines and infrastructure, given the high expectations for CCS technology in both the energy supply and industry sectors, the appropriate layout of the pipeline network and commercial promotion is essential for CCS popularization.

**Transition costs and co-benefits**. Understanding that the energy transition will create a huge need for investment and technological innovation that drives economic growth, but also brings economic burden and welfare loss. The aggregate results show that the energy supply sector requires an investment of 4.9–7.8 trillion US dollars during 2020–2050 to kickstart the zero-carbon transition, representing an increase of at least 65% over that in the NDC scenario (Fig. 8). The 2020–2030 investment will increase, by 43% (over a 21–67% range), if emissions in China peak in 2025 rather than in 2030, but the welfare loss of more than five times the new investment can be avoided (Fig. 6g and Supplementary Fig. 17). Furthermore, if emissions in China peak in 2020, for the 25th–75th percentile cases, the energy supply sector requires a 99–162 billion US dollars investment increase in the next decade compared to that in PEAK25, which is still a huge jump despite investment reduction in coal-fired power.

Considering the whole energy system, on average, the cumulative GDP loss for 2020–2050 due to energy system investment, maintenance and operation costs in the mitigation scenarios is 3.3–3.6%, compared to 3.2% for the NDC scenario. Significant reductions in operating costs offset most of the increasing investment costs. The small increase in total energy system costs also suggests that China's energy transition is achievable.

As illustrated in Fig. 6g, the welfare loss reflects the utility changes of end users. Similar values of welfare loss are observed for the PEAK20 (0.4–1.7 trillion US dollars) and PEAK25 (0.4–2.3 trillion US dollars), and these values are significantly lower than the 1.9 trillion US dollars (over a 0.7–3.5 trillion US dollar range) for PEAK30. Such high growth in welfare loss for PEAK30 is due to insufficient emission reductions before 2030 compared to those in the other two scenarios, leading to excessive pressure to meet the national cumulative carbon budget, requiring not only the considerable deployment of advanced technologies but also a massive decline in the energy service demand. Combining the results of investment and welfare losses, we find that an earlier peak aids in achieving a complete energy transition and reduces the transition burden, resulting in a win-win for both economic growth and climate governance.

We find that the marginal abatement cost (narrowly defined as the social cost of carbon), is subject to significant uncertainty (Fig. 9a, b). The interquartile range of the marginal abatement cost is 89–251 US dollars in 2035 and 174–492 US dollars in 2050 under the PEAK30 scenario but is below 151 and 258 US dollars in 2050 for PEAK20 and PEAK25, respectively. The PEAK20 scenario has the lowest marginal abatement cost in 2050, mainly because the substantial effort in the short term reduces the pressure for subsequent mitigation. Figure 9c shows that the marginal abatement cost increases exponentially with the abatement rate. The marginal abatement cost is often equivalent to the optimal carbon price, and therefore policymakers can observe the carbon price needed to achieve the corresponding level of emission reductions in different periods. As shown in Supplementary Fig. 18, the PEAK30 scenario is significantly more sensitive to future technological potential and costs than the other

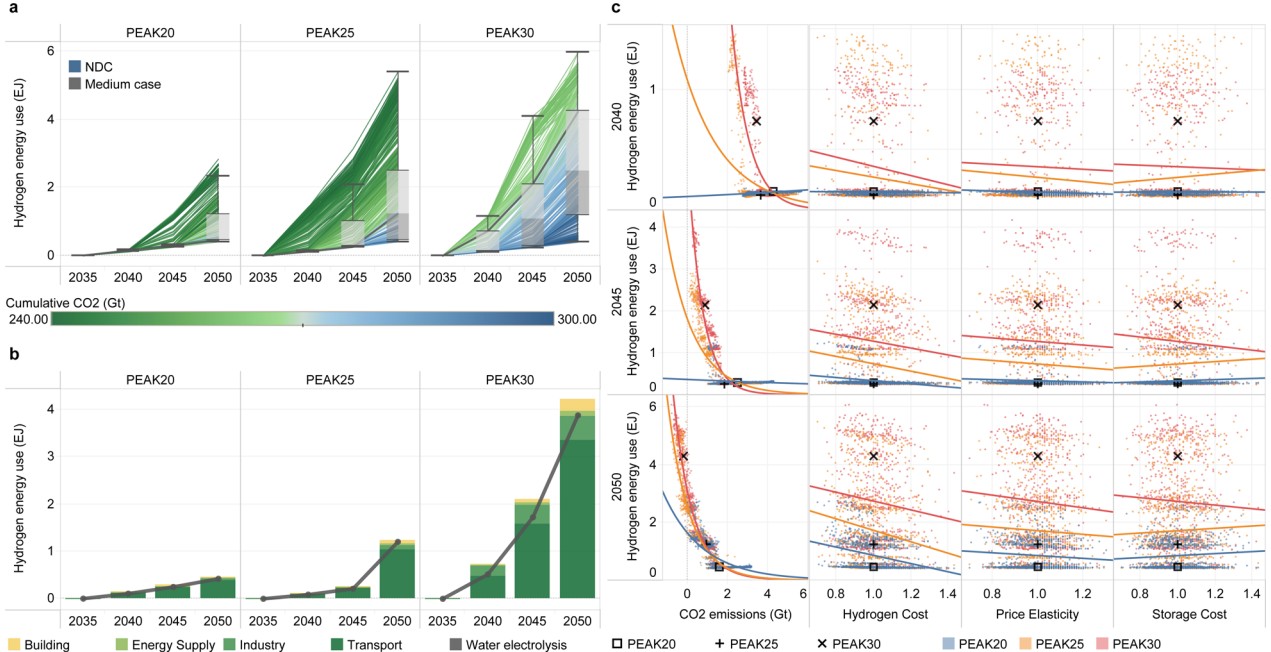

**Fig. 7 Uncertainty analysis of hydrogen energy. a** shows the development of hydrogen energy (unit: EJ). The box plot shows the first quantile, intermediate range (IQR), and third quantile of all the results, where the data range within 1.5 times the IQR is denoted with whiskers. The thick blue line represents the pathway of the NDC scenario, and the thick gray line represents the pathway for the intermediate case in each scenario. The divergent color from blue to green reflects the increasing stringency of the cumulative carbon budget. The cumulative $CO_2$ parameter corresponds to the absolute value of China's cumulative carbon budget for 2010–2050. **b** presents a decomposition of the different uses of hydrogen energy (stacked bar chart) and the production of hydrogen from water electrolysis (black line) for the intermediate cases. **c** represents the relationship between the development of the hydrogen energy and uncertain input parameters and $CO_2$ emissions. Three parameters with significant impacts among the fourteen uncertain input parameters are shown for each technology. The remaining results are shown in Supplementary Fig. 12. The variables (except for the $CO_2$ emissions) represent multiples of the intermediate cases for each scenario. The intermediate cases of PEAK20, PEAK25, and PEAK30 are denoted with square, plus, and multiplication signs, respectively.

two scenarios. Cost reduction in CCS and hydrogen energy, can significantly reduce the marginal abatement costs in 2050, reflecting the importance of technological innovation in reducing the policy costs of climate governance.

Energy transitions have created synergies in local air pollutant reduction. Even without further air pollutant control measures and relying on carbon emission reductions alone, PEAK25 is associated with decreases in the emissions of sulfur dioxide ($SO_2$), nitrogen oxide ($NO_X$), particulate matter with an aerodynamic diameter no greater than 10 μm ($PM_{10}$) and particulate matter with an aerodynamic diameter no greater than 2.5 μm ($PM_{2.5}$) by 14–27%, 18–26%, 13–23% and 12–22%, respectively, in 2030 compared to the present levels, while the corresponding reductions in PEAK30 are only 6%, 12%, 7% and 7%. In 2050, local air pollutant emissions for almost all scenarios fall to only one-fifth of the current values (Fig. 10). The results are very consistent with the previous study's result when carbon neutrality is achieved[38].

## Discussion

A probabilistic representation of technology choice and fuel switching dynamics for the energy transition shows that China's carbon neutrality goal is not impossible but requires a cross-sectoral, collaborative effort. Although the future of policy and technology development are full of uncertainties, some important consensus can be obtained from this study. Most importantly, decisive action and clear goals enable us to realize carbon neutrality. In particular, clean energy supply, low-carbon energy consumption, and green energy investment go hand in hand on the road to carbon neutrality.

Net-zero emission targets should not distract from the urgent need for near-term emission reductions. Understanding that energy system transition cannot be accomplished overnight, decisive actions facilitate an earlier emission decline, thus buying time for China's transition to a low-carbon economy at the domestic level. If China's $CO_2$ peak occurs in 2025 instead of 2030, welfare losses will be reduced by at least 50%, marginal abatement costs in 2050 will be kept within reasonable limits and heavy reliance on carbon removal technologies will be avoided. At the international level, these actions lay a foundation for China to make an ambitious contribution to mitigating global climate change. With a peak in 2030, it is almost impossible for China to achieve cumulative emissions below 250 Gt in 2010–2050, meaning that either the 1.5-degree target called for in many studies will be difficult to achieve or China will have to generate significant negative emissions for a long time in the second half of this century. Recently, more than 150 parties, including China, submitted their new or updated NDCs, and most have raised their targets to increase their ambition to address climate change. For China, an enhanced NDC would boost climate mitigation while facilitating a domestic transition. It is wise for China to implement the enhanced NDC, aiming to reach an earlier and lower emission peak. Achieving a carbon peak near 2025 with a peak ~10.3 $GtCO_2$ and declining to ~9 $GtCO_2$ in 2030, is the possessive choice to combine near-term and long-term transition pressures.

Our study suggests that a concrete and transparent total carbon emission control mechanism can largely reduce uncertainty. Policymakers need to consider setting total emission control targets and cumulative emission targets rather than carbon emission intensity reduction targets alone to encourage

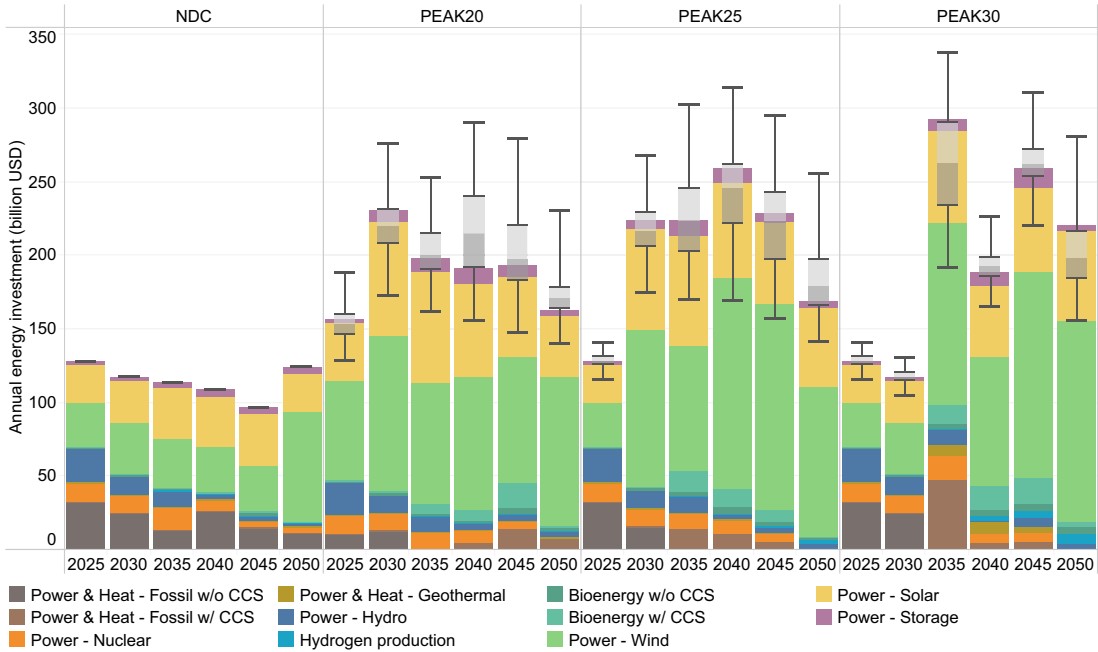

**Fig. 8 Annual investment in energy supply (unit: billion US dollars).** The annual investment in biomass utilization, hydrogen production, energy storage, fossil fuel power generation and heating, solar power generation, wind power generation, and other renewable energy power generation from 2020 to 2050 for the intermediate case in each scenario is shown in the stacked bar chart. The box plot represents the uncertainty range of annual total energy supply investment. The box plot shows the first quantile, intermediate range (IQR), and third quantile of all the results, where the data range within 1.5 times the IQR is denoted with whiskers. The uncertainty in the cumulative investment for each type of technology for 2010–2050 is presented in Supplementary Fig. 17. In this figure, w/ CCS means that this technology is equipped with carbon capture and storage, while w/o CCS means that it is not equipped.

progressive emission reductions. Net-zero emission targets alone, while cumulative emissions remain highly uncertain, will ultimately lead to large differences in emission reduction pathways, technology choices and transition costs. For China, an appropriate total carbon emission control target should be established in conjunction with the announced carbon neutrality target in a way that does not impose an excessive burden on society, appropriately carries international responsibility and contributes to the 1.5-degree target.

Key technologies for achieving carbon neutrality have not been fully developed, and it is important to cultivate diverse technologies and build a portfolio of carbon-neutral technology reserves. Innovation in renewables and advanced technologies is a critical backbone of the energy transition. Rapid reductions in PV and wind power costs have already changed the landscape for addressing climate change. Considering the massive demand for variable renewable energy, further technological progress in wind, solar, and energy storage will powerfully impact the transition cost. Hydrogen, biomass, and CCS technologies, which in the past have often been used as back-up resources, will be important pillars of carbon neutrality due to their indispensable roles in a net-zero and carbon-negative world; hence, there is an urgent need for sustained R&D investments and industrial pilots for these high-cost technologies.

While the energy supply sector has considerable potential for emission reduction with the popularization of renewable energy and the elimination of coal in the near term, other sectors still need to take action without delay. The continued elimination of energy-intensive industries and an increase in the value of products are needed in the industry sector. Fuel substitution in the building and transport sectors fundamentally alters energy use patterns and requires public acceptance. Increased public awareness of carbon neutrality helps to consciously adopt a low-carbon lifestyle and reduce energy consumption and the carbon footprint.

The carbon neutrality goal requires us to examine the current financial system and shift investments from carbon-intensive assets to green assets. Investment is the vane of future development. The results have clearly indicated that the existing thermal power is associated with a high risk of capital stranding. Hence, we suggest a moratorium on new thermal power, and thermal power plants under construction should be capture ready. In the context of carbon neutrality, it is imperative to reduce fossil energy investments, support renewable energy expansion, and promote international cooperation. The launch of China's national emission trading market further reasonably shares the mitigation cost and encourages decarbonization at the enterprise level.

Overall, the significant uncertainties in the emission peak time, technological development and cumulative carbon budget lead to multiple pathways for China towards carbon neutrality. In our study, robust conclusions regarding emission pathways, energy transitions, consumer behavior shifts, transition costs and co-benefits are obtained for policymakers to deploy national mitigation strategies. At present, over 130 countries have announced their carbon neutrality goals, which is a major step forward; however, more attention should be given to near-term actions, emission reduction pathways towards carbon neutrality, and the remaining national cumulative carbon budget required.

## Methods

**China-TIMES model.** The China-The Integrated MARKAL-EFOM System (China-TIMES), a dynamic linear programming energy system optimization model, was developed for 5-year intervals extending from 2010 to 2050 on the basis of the China MARKAL model[39–41]. The base year for the China-TIMES model is 2015. The model uses 5% as the discount rate. The model incorporates the full range of energy processes including exploitation, conversion, transmission,

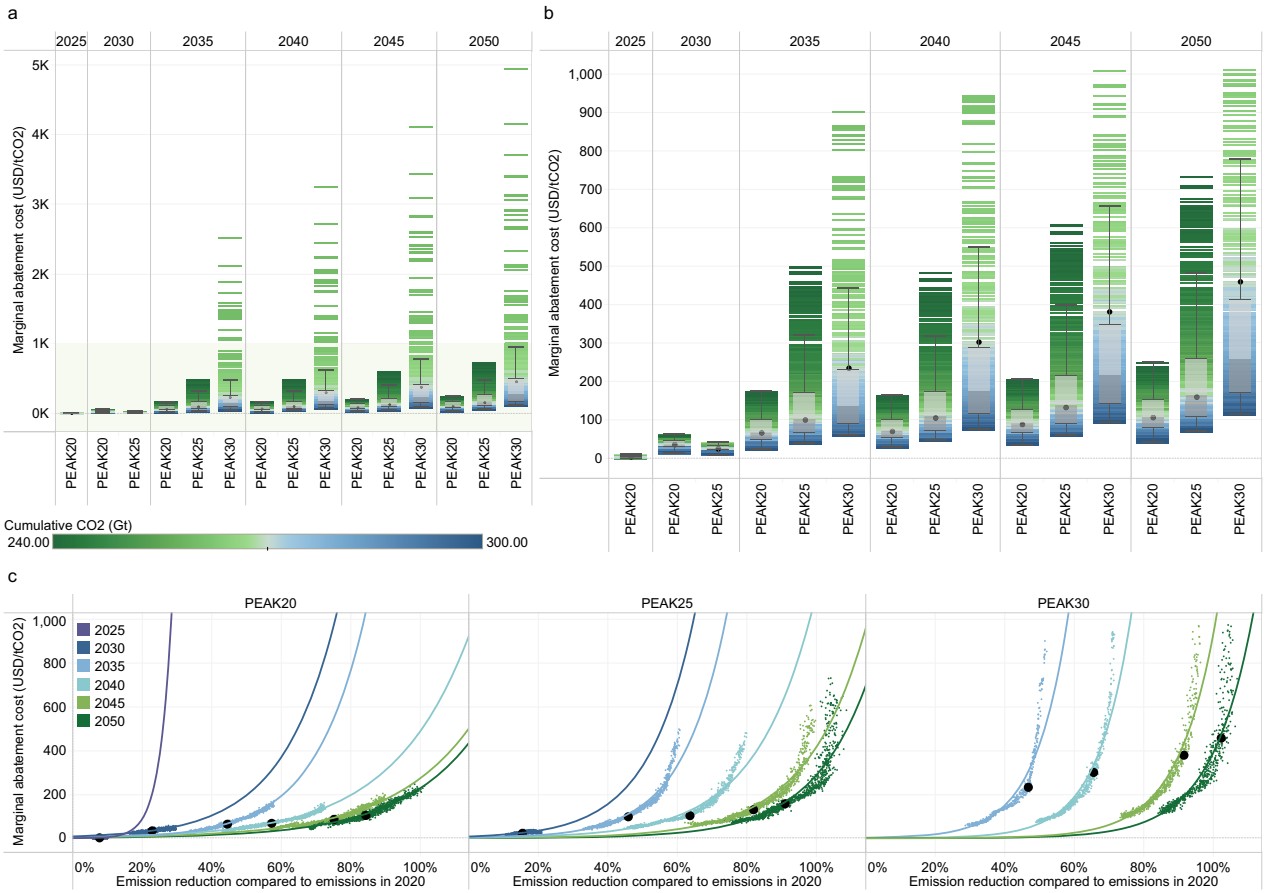

**Fig. 9 Marginal abatement cost under different scenarios (unit: US dollars/tCO₂). a** represents the marginal abatement cost of all feasible cases.
**b** enlarges the concentrated data in panel **a** (green shadow) corresponding to all cases with carbon prices below 1,013 US dollars per ton CO₂ by 2050
(90% of the total feasible cases for PEAK30). **c** Relationship between the marginal abatement cost and the CO₂ emission reduction in reference to
emissions in 2020 (as fitted by an exponential function). The black points represent the marginal abatement costs for the intermediate cases in each
scenario. The box plot shows the first quantile, intermediate range (IQR), and third quantile of all the results, where the data range within 1.5 times the IQR
is denoted with whiskers. The divergent color from blue to green reflects the increasing stringency of the cumulative carbon budget. The cumulative CO₂
parameter corresponds to the absolute value of China's cumulative carbon budget for 2010–2050. Supplementary Fig. 18 shows the relationships between
the marginal abatement cost and other uncertain parameters.

distribution and end-use. Over 500 existing and advanced energy supply and demand technologies are introduced in the model. Three demand sectors, industry[19–21], building (divided into commercial, urban residential and rural residential)[22] and transport[23], are considered and further divided into over 40 subsectors. Daytime and night-time are modeled separately, and intraday and interseason load differences are fully considered.

For the power sector, we consider different fuel types (coal, oil, gas, wind, solar, nuclear, hydropower, marine, biomass and geothermal). Different thermal power types have different cost and efficiency settings depending on the technology category (ultra-supercritical, supercritical, subcritical, ultra-high pressure, circulating fluidized bed, pressurized fluidized bed, gasification combined cycle, etc.) and cooling method (air cooling, direct water cooling, or circulating water cooling). For renewables, the model includes hydropower, centralized PV, building integrated PV (BIPV), concentrating solar power (CSP), offshore and onshore wind power, biomass energy, geothermal energy, tidal energy, and wave energy at different scales and development levels. In this version of China-TIMES, we refine the description of biomass power generation, which includes biomass gasification, direct combustion, and co-combustion with coal. For thermal and biomass power generation, we consider various CCS technologies available for existing unit retrofits and new construction. The installed capacity, efficiency, and capacity factor of each type of power generation technology are set according to real data from the China Electricity Council.

Stock-based material flow analysis approach, discrete choices method, Gompertz model etc. are used to project energy service demands for the subsectors according to given socioeconomic development scenarios. The socioeconomic driver settings such as the gross domestic product (GDP), population, urbanization level, and industrial structure are listed in Supplementary Table 1. The projected energy service demands for more than 40 end-use subsectors are met by abundant

energy end-use demand technologies, thereby driving energy system optimization. In the industry sector we focused on the production of steel, cement, glass, nonferrous metals, paper, and chemicals (ethylene, sodium hydroxide, sodium carbonate and ammonia) and created process-level descriptions. In addition to emissions from fuel combustion, we considered emissions from the cement production process, which are generated by the decomposition of raw materials. The future penetration of hydrogen, electrical energy and CCS in the industry sector is fully considered. For the building sector, space heating, cooling, cooking, water heating, lighting, and electric appliances are considered. Taking space heating as an example, in addition to modeling different heating methods and heating needs for China's five climate zones, energy efficient building standards for different periods are also considered. Future clean and efficient technologies, such as solar thermal, geothermal, and air source heat pumps, are also considered. The transport sector is divided into passenger and freight transport, and further split into aviation, navigation, railway, highway (urban, rural, and intercity), and pipeline (for freight); additionally, many different types of vehicles are considered. The model portrays the role of fuel economy improvements, fuel substitution, and travel mode shifts in a balanced way.

Energy storage technologies (including pumped hydro storage, compressed air storage, flywheel storage, and various types of electrochemical storage) can bridge the gap between the energy supply and demand at different times. In this study, we focus on intraday energy storage, but the different characteristics of energy storage technologies in the model and the many demand-side management tools (dynamic hydrogen production, energy demand balancing between seasons, etc.) are able to provide a good understanding of energy storage demand.

Local air pollutants[24,25] such as SO₂, NOₓ, PM₁₀, PM₂.₅ as well as energy-related water consumption[26] can also be simulated with the China-TIMES model. The emission factors for CO₂ are set according to IPCC guidelines[42] and remain

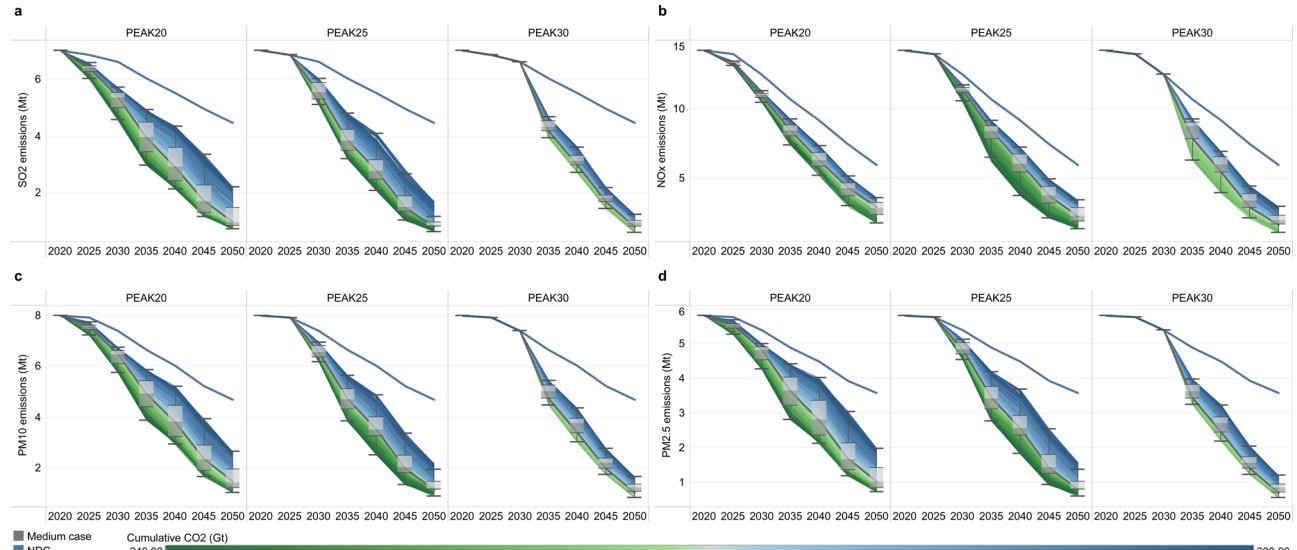

**Fig. 10 Synergistic benefit of the CO₂ emission reduction on air pollutant control (unit: Mt). a–d** shows sulfur dioxide ($SO_2$), nitrogen oxide ($NO_X$), particulate matter with an aerodynamic diameter no greater than 10 μm ($PM_{10}$) and particulate matter with an aerodynamic diameter no greater than 2.5 μm ($PM_{2.5}$) emissions, respectively. The control measures of $SO_2$, $NO_X$, $PM_{10}$ and $PM_{2.5}$ are based on the current levels; that is, only considering the synergy of $CO_2$ emission reduction, not considering the effects of subsequent local air pollutant control measures. The box plot shows the first quantile, intermediate range (IQR), and third quantile of all the results, where the data range within 1.5 times the IQR is denoted with whiskers. The thick blue line represents the pathway for the NDC scenario, and the thick gray line represents the pathway for the intermediate case in each scenario. The divergent color from blue to green reflects the increasing stringency of the cumulative carbon budget. The cumulative $CO_2$ parameter corresponds to the absolute value of China's cumulative carbon budget for 2010–2050.

constant over time. The emission factors for local air pollutants are calibrated with official statistics and kept constant over time to correctly reflect the synergistic effects of $CO_2$ emission reductions.

**China-TIMES-MCA model**. The China-TIMES-MCA model (Fig. 1) implements coupling with Monte Carlo analysis (MCA) on the basis of the China-TIMES model to perform large-scale uncertainty analysis. We use the Latin hypercube sampling (LHS) method[43,44] to replace the random sampling method used in the classical Monte Carlo simulation. LHS is a stratified sampling method that approximates random sampling. Multivariate parameters are sampled using a relatively small number of sampling points to resolve the aggregation problem that can occur in random sampling and ensure that low-probability events are accurately represented in the model.

According to the nature of different parameters, we establish the uniform distribution, lognormal distribution, and normal distribution, while ensuring that the median of each distribution is 1. Different parameters are randomly generated in the same scenario. The sample probability distributions and statistics summary are shown in Supplementary Table 2. The parameters of cases with the same order number are identical for different scenarios (such as the 1st case for PEAK20 and the 1st case for PEAK25), to effectively compare the differences among scenarios.

**Generation of uncertain input parameters**. We use predetermined probability distributions to model the factors that can significantly impact China's energy system transformation and climate change mitigation. The fourteen selected key parameters are the cumulative carbon budget (Carbon Budget), technological cost of BECCS (BECCS Cost), utilizable biomass resource potential (Bio Cap), PV power cost (Solar Cost), PV power economic installed capacity (Solar Cap), wind power cost (Wind Cost), wind power economic installed capacity (Wind Cap), energy storage cost (Storage Cost), nuclear power cost (Nuclear Cost), nuclear power economic installed capacity (Nuclear Cap), thermal power with CCS cost (Thermal CCS Cost), industry CCS cost (Industry CCS Cost), hydrogen production, storage and transportation cost (Hydrogen Cost), and price elasticity of energy service demands (Price Elasticity).

Except for the cumulative carbon budget, the parameters of the intermediate case are based on the IEA reports, industry statistics, and existing literature (see Supplementary Table 3)[2,45–53]. The parameters for other cases are generated using adjustment multipliers based on those used for the intermediate case. The cumulative carbon budget is an important indicator of a country's contribution to global climate change mitigation. Various studies have shown that considerable uncertainty remains in the remaining carbon budget[54,55]. Since China's carbon budget varies significantly under different allocation principles, we set an uncertain input parameter to reflect the efforts of China's emission reduction. The carbon budget parameter is the absolute value of China's cumulative carbon budget from

the beginning of 2010 to the end of 2050. The carbon budget parameter considers only energy-related $CO_2$ emissions and does not include LULUCF emissions. We integrate various fairness principles to construct the carbon budget parameter, which is centered at 270 Gt and uniformly distributed from 240 to 300 Gt, which covers the median carbon quotas for achieving the 2- and 1.5-degree targets[27,56–59].

**Scenario design and simulation**. A deterministic reference scenario and three groups of uncertain mitigation scenarios of which 1000 independent cases each are included in this study. The reference for the mitigation scenarios is NDC, which includes China's NDCs and climate-related policies published before 2020. The three groups of mitigation scenarios are all oriented towards China's carbon neutrality goal and represent different emission peak times and time at which a carbon neutrality-oriented emission reduction trajectory is observed through from 2020 (PEAK20), 2025 (PEAK25), and 2030 (PEAK30). The same socioeconomic assumptions are used in all scenarios to generate the energy service demand for end-use sectors and subsectors in the China-TIMES-MCA model. This model integrates resource availability, technology cost and technology availability on both the supply and demand sides to seek the lowest cost technology and fuel mix to meet the energy service demand and carbon constraints. The price elasticity of the energy service demand for each end-use demand subsector is introduced to consider emission reductions based on changes in the production mode and consumption pattern. The choice of price elasticity of the intermediate case is referenced from previous study[60]. The early retirement of units in energy-intensive industry sectors, and thermal power and heat generation units without CCS is allowed in the model. On a PC with an Intel i7-9700, 32 GB RAM, and 1 TB SSD, a round of optimization using CPLEX takes 6–8 h, and the performance of the computer and solver can have a significant impact on the computing time. For the PEAK30 scenario, 325 out of 1000 cases are not included because optimization is infeasible.

**Reporting summary**. Further information on research design is available in the Nature Research Reporting Summary linked to this article.

## Data availability

The energy system portion of China-TIMES-MCA was calibrated with the China Energy Statistics Yearbook, which is a public publication. Local air pollutants were calibrated according to the "Second National Pollution Source Census Bulletin", which can be viewed on the website of the Ministry of Ecology and Environment (http://www.mee.gov.cn/xxgk2018/xxgk/xxgk01/202006/t20200610_783547.html). The input data for the key uncertain parameters discussed in the text are disclosed in the Supplementary Information. China-TIMES-MCA model output data generated in this study are provided with this paper in the Source data. Additional data that support the

findings of this study are available from the corresponding author upon reasonable request. Source data are provided with this paper.

## Code availability

The code and documentation of the TIMES model are open source and available on GitHub (https://github.com/etsap-TIMES/TIMES_model). Latin hypercube sampling used for generating uncertain input parameters, model data transfer, model input file generation and some visualization tasks, we used FORTRAN, MATLAB and R, the code can be found at https://doi.org/10.5281/zenodo.5717886[61]. For other visualization tasks, we used Tableau and Visio, with no programming involved.

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

## Acknowledgements

This study was supported by the National Natural Science Foundation of China (71690243 and 51861135102 to W.C.) and the Ministry of Science and Technology of the People's Republic of China (2018YFC1509006 to W.C.).

## Author contributions

S.Z. and W.C. conceived the study, performed the analysis, and contributed to writing the paper.

## Competing interests

The authors declare no competing interests.
