## [Peer Review File · Nature Communications]

Peer review first comments –

Reviewer #1 (Remarks to the Author):

Reviewer's comments to COMMS-21-14011-T

Summary of review: The paper investigated a very important policy issue in China's climate policy: how to advance policy discussions under uncertainties. The authors made enormous efforts to put together the model, and presented the results and findings in a beautifully virtualized form. The paper fills a critical gap and I'm excited about the research.

A few and questions and comments for the paper to consider:

1. I assume the model is annually based, I'm not clear how the storage technologies are configured with variable renewables at annual based data, and how the model simulates the need for storage?
2. I'm surprised that nuclear capacity/costs are not in the key uncertainty parameter. In general, there are three main pathways in the discussion: 1) nuclear; 2) coal + ccs; and 3) renewable + storage. Each represents a combination of technology and investment needs, and different policy options/implications. I do not mean the authors have to do nuclear, but just curious how the author weighs in when choosing the 8 key uncertain parameters.
3. I understand China-TIMES is a massive model, and this paper focus on MCA, however, it would be good the paper can add a session to introduce more about the model, and direct the reader material/manual/paper that has detailed documentation of the formula, parameters, and sources of the data, etc. What's the base year? What discount does the model use? In addition, what software, environment, and resources are needed to run China-TIMES-MCA, will the model and data be open-source? Those could help the readers understand better of the model.
4. Name in Supplementary Fig. 1 China-TIMES-MCA structure: I suggest using "4 Deterministic Scenarios" and "3000 Uncertain Cases", so to differentiate the different focuses. I'm not clear about how the 3000 cases are generated? How the 8 key parameters' probability distribution change over the different cases? Random? Simultaneous? Combination?
5. What emission factors does the paper use when calculating emissions? Assuming the same emission factors over time? Or different ones?
6. Given this MCA practice, I'm looking for some well-summarized conclusions that from the simulation. Which scenario do the authors promote? What are the key uncertainties/factors for policy making? What are
7. Other minor suggestions:
 - a. Fig 2. Change the order of technology in legends to the same order as in the stacked area chart so to make it more readable. The unit of power installed capacity is TW, not GW.
 - b. Supplementary Table 1, are 2010, 2015, 2020 historical actual data or assumption data? It might be useful to note the differences if are assumption, and could be used to compare the modeling behavior and performance.
 - c. Supplementary Table 3. What are the cost assumptions of conventional technologies: coal, gas, nuclear, hydro technologies? Coal combustion with CCS cost seems too low? Could you please provide the sources/rationales of those numbers?

Reviewer #2 (Remarks to the Author):

This paper conducts a Monte Carlo analysis on several key parameters using the China-TIMES model. Several thousand scenarios are generated, and basic statistics are carried out on the scenario results, in order to identify important insights for the evolution of China's energy mix and emissions out to 2050.

The methodology employed in this study is solid - for the most part. The China-TIMES model is well-documented and well-respected in the literature. The novel methodological aspect of this study is the use of Monte Carlo analysis, which the authors perform in a straightforward way. The one issue I have with the study approach, however, is that the number of uncertain parameters that are varied in the Monte Carlo analysis are relatively few (8 in total), and pretty much all of

them are focused on power sector technologies (Supplementary Table 2). It would be important to also include other uncertain parameters in the Monte Carlo analysis, such as the techno-economics of various end-use sector technologies, or perhaps the price elasticities on service demands.

The results of the paper are very important, not just for China but for the world at large. The paper reports a huge amount of results, both in text and figures. However, the paper is not told as a story, but is rather diagnostic in its presentation of results. This will lose the attention of many readers very quickly. It would be preferable to highlight the importance of certain results as they are described, perhaps mentioning when things are counter-intuitive and/or how they compare to other recent studies found in the literature (not just for China).

Moreover, the paper is missing some punch lines. What are the main messages that the reader should take away? Which uncertainties matter more than others? Could the authors even do something more sophisticated in their analysis of scenario results – something like Robust Decision Making / Scenario Discover (Lempert et al. papers) – in order to identify the most impactful uncertainties? I would like to highlight that the insight on p. 12 (also in the abstract) about nearer-term peaking resulting in lower welfare loss and less reliance on carbon dioxide removal is a crucially important finding.

The quality of written English in the paper is very good, and the presentation and organization work very well.

Some more specific comments below...

Page 4 => It would be helpful to have a table (or diagram) here that clearly explains the scenario design.

Page 5, first paragraph => These are fossil fuel and industrial sector emissions only, right? No land use emissions included here? It would be helpful to state this explicitly at the start, so that there is no confusion on the part of the reader.

Page 5, second paragraph => Please state clearly whether the 240-300 GtCO₂ carbon budget is for China or for the world as a whole.

Page 5, second paragraph => There is a comma (,) before the word 'Note'. I think it should instead be a period (.).

Page 10, middle of page => According to my understanding, these more fundamental changes in consumer behavior and lifestyle (remote office work, online shopping, sharing economy) are outside the scope of the price-induced demand responses modeled (i.e., the price elasticities do not cover those more major changes in demand). Some further explanation is needed on this point, in order for the reader to understand.

Page 11, middle of page => Are the investment numbers cumulative, undiscounted values? Over which timeframe (2010-2050)?

Page 17, top of page => Please be specific that the carbon budget is in terms of Gtons CO₂ from fossil fuel and industrial emissions (excluding land-use emissions), if I am correct.

Page 17, bottom of page => I note that demand reduction is possible via price elasticities; however, vastly different demand trajectories have not been created (e.g., following different SSPs). The authors should state this explicitly, referring to Supplementary Table 1, either here or on p. 15.

Reviewer #3 (Remarks to the Author):

This paper uses numerous scenarios generated by Monte Carlo method for several variables with pre-defined distributions. It is well written, but I have some significant concerns on the analysis in the paper.

First, the time horizon used for the analysis of the paper is 2010-2050, while China's carbon neutrality target year is 2060. In this sense, it will be much better to include both peak year and neutrality year in scenario design to deliver good insights on China's transition pathways.

Second, I do not see obvious advantages of Monte Carlo method in such a type of analysis than well-designed much less sensitivity analysis. The authors do not explain in the SI on why and how they setup the distributions and relevant values for the 8 so-called key parameters. This is very crucial for the entire analysis and should be well described and explained.

Third, I do not think solar and wind potential are that important to be included in the 8 selected key parameters. The ultra-high voltage (UHV) technologies currently widely used in China could transmit large-volume remote PV and wind power in the north and west of China to demand centers in China's east and south, and the UHV network is planned to be largely expanded in the future. In this sense, as the model used by the authors has no sub-national resolution, I do not see any significant reasons to heavily analyze the solar and PV potential here. Moreover, I think that hydrogen use should be a key technology for deep decarbonization and for achieving carbon neutrality from carbon peak within a very short time period (like 30-40 years) in China, but it is not included in the currently designed Monte Carlo simulation.

Forth, I do not see a tight bond between the main analysis part of the paper and the part of discussion and policy insight. The current discussion part is rather general. The authors may want to revise the discussion part to well reflect their key and relevant findings in the analysis part.

Response to Reviewers

NCOMMS-21-14011A

October 1, 2021

Reviewer #1

General comment:

The paper investigated a very important policy issue in China's climate policy: how to advance policy discussions under uncertainties. The authors made enormous efforts to put together the model, and presented the results and findings in a beautifully virtualized form. The paper fills a critical gap and I'm excited about the research. A few and questions and comments for the paper to consider.

Response:

Thank you for your valuable comments, as they are insightful and lead to the improvement of our work.

Comment 1.1:

I assume the model is annually based, I'm not clear how the storage technologies are configured with variable renewables at annual based data, and how the model simulates the need for storage?

Response:

Thank you for your raising this issue. In the model, we have mainly considered the need for energy shifting energy storage. And in the vast majority of them, they are used for intraday energy storage. According to relevant studies, energy storage for this purpose accounts for about 70% or more of the total energy storage. Hence, we modeled daytime and nighttime energy supply technologies and energy service demand independently for China-TIMES-MCA: power generation technologies operate differently during daytime and nighttime (e.g., PV cannot generate electricity at night, wind power usually generates more electricity at night than during the day, etc.); energy service demand also differs day and night (e.g., electric vehicle charging is more concentrated at night, etc.). Since there are differences in energy supply and demand between day and night, energy storage technology can be deployed in this area to achieve intra-day balancing of electrical energy. Inter-period storage of energy (across seasons, years) is also addressed in our modeling framework. Energy storage technologies such as compressed air storage and pumped hydro storage as well as demand-side management tools such as timely hydrogen production and inter-seasonal energy demand balancing are considered in the model.

Changes/additions to the manuscript:

- **Manuscript, Page 22-23, Line 411-418**

Abundant energy end-use technologies translate energy service demand into end-use energy demand, thereby driving energy supply optimization. Energy storage technologies including pumped hydro storage, compressed air storage, and electrochemical storage can bridge the gap

between energy demand and supply at different times. In this study, we focus on intra-day energy storage, but the different characteristics of energy storage technologies in the model and the rich demand-side management tools (dynamic hydrogen production, energy demand balancing between seasons, etc.) are able to provide a good understanding of energy storage demand.

Comment 1.2:

I'm surprised that nuclear capacity/costs are not in the key uncertainty parameter. In general, there are three main pathways in the discussion: 1) nuclear; 2) coal + ccs; and 3) renewable + storage. Each represents a combination of technology and investment needs, and different policy options/implications. I do not mean the authors have to do nuclear, but just curious how the author weighs in when choosing the 8 key uncertain parameters.

Response:

Thank you for your valuable comments. There is no doubt that nuclear, renewables and CCS are all essential technologies for China to achieve deep decarbonization. We have also detailed modeling of them all. In the revised version of our paper, we have added 6 new key uncertain parameters including *thermal power with CCS cost*, *nuclear power cost* and *nuclear power economic installed capacity* and re-simulated the scenarios.

As the cost of renewable energy and energy storage has fallen by 90% in the last 10 years, the competitiveness of renewable energy has increased significantly, and there is a lot of space and uncertainty in the future movement of costs, thus becoming the focus of our paper. Since a small number of hard-to-abate emissions need to be offset by negative emissions, the cost of BECCS technologies and the potential for economic utilization of biomass resources are set as key uncertainty parameters.

As for nuclear power, the investment cost of third-generation reactor is significantly higher than that of second-generation due to safety concerns, while future cost reductions are limited. In addition, in China's 14th Five-Year Plan, the pre-research plan for inland nuclear power plants was not highlighted while it was mentioned in the 13th Five-Year Plan, so the future development of inland nuclear power might be constrained. In the model, currently common pressurized water reactor (PWR) technologies, as well as higher safety fourth-generation reactor types which can be built inland (e.g., high-temperature gas-cooled reactors (HTR) and small reactors) are considered. But after 2035, nuclear power will not only be a safety concern, but it will also have no cost advantage over PV with energy storage. The development of coastal nuclear power is mainly constrained by the limited location of available plants, while inland nuclear power is more costly and difficult to develop in large quantities. The results also show that nuclear power will be deployed more in the future when PV costs fall less than expected.

As for thermal power with CCS, post-combustion capture units, oxygen-enriched combustion units and IGCC with CCS units of different capacity levels are modeled separately in the model. Retrofitting coal-fired power plants with CCS can transform them into a relatively low-carbon generation technology that can delay the premature retirement of thermal power due to carbon

constraints. This is one way to address the stranding of thermal power assets. In addition to this, thermal power can also reduce asset stranding losses by providing ancillary services of backup and frequency regulation. The direct conversion of coal-fired power to BECCS plants, which offsets other emissions, is a more expensive but efficient method. In the model, these technology routes are cost-competitive to obtain optimal results in terms of installed power and generation capacity. The development of thermal power with CCS competes more with other pathways in the PEAK20 and PEAK25 scenarios, while in the PEAK30 scenario, all low-carbon technologies must have a very engaged rate of penetration and scale of installed capacity.

Changes/additions to the manuscript:

- **Manuscript, Page 11, Line 180-188**

Despite being a stop-gap measure, thermal power with CCS also gains growth opportunities beyond 2035, providing power system flexibility and some heating needs. The development of thermal power with CCS units shows a smooth gradient of increasing capacity with decreasing cumulative carbon budget. For the 25th-75th percentile cases, thermal power with CCS capacity reaches 151 GW (over a 130-169 GW range) for PEAK20 and 154 GW (over a 150-202 GW range) for PEAK25 in 2050, most of which is retrofitted from existing thermal power. For the PEAK30 scenario, 225-249 GW of thermal power with CCS is needed in 2050, and most of it will be gas-fired power with CCS to further reduce emissions.

- **Manuscript, Page 11-12, Line 189-202**

Nuclear power, with its high capacity factor and supply reliability, will grow rapidly in the next 20 years as the ballast of the power system. There are 31 coastal sites and 46 inland sites available for nuclear power construction, in other words, about 200 GW and 250 GW of coastal and inland nuclear power can be built³⁵. Limited site location is a major constraint to nuclear power development (Fig. 4e). For the intermediate cases of all scenarios, the installed capacity of coastal nuclear power reaches 176 GW in 2040, 3.5 times higher than in 2020. By 2050, the installed capacity rises slightly to 190 GW, and the sites available for nuclear power construction along the coast are largely built out. Variation between cases comes from the choice of nuclear unit capacity at each site. The results also show that inland nuclear power has the potential to reach a modest scale of construction (less than 150 GW) only in the face of extremely stringent cumulative emission targets, hindered renewable energy development, and very rapid declines in nuclear power costs in PEAK30 scenario. Meanwhile, in the latest China's 14th Five-Year Plan, the expression for inland nuclear power development has been deleted. The future advancement of inland nuclear power may be inhibited.

- **Manuscript, Page 24, Line 442-450**

The fourteen selected key parameters are the technological cost of BECCS (**BECCS Cost**), utilizable biomass resource potential (**Bio Cap**), PV power cost (**Solar Cost**), PV power economic installed capacity (**Solar Cap**), wind power cost (**Wind Cost**), wind power economic installed capacity (**Wind Cap**), energy storage cost (**Storage Cost**), nuclear power cost (**Nuclear Cost**), nuclear power economic installed capacity (**Nuclear Cap**), thermal power with CCS cost (**Thermal CCS Cost**), industry CCS cost (**Industry CCS Cost**), hydrogen production, storage and

transportation cost (**Hydrogen Cost**), and price elasticity of energy service demands (**Price Elasticity**).

New References in Manuscript

35. Xiao, X. J. & Jiang, K. J. China's nuclear power under the global 1.5 degrees C target: Preliminary feasibility study and prospects. *Adv. Clim. Chang. Res.* **9**, 138-143 (2018).

● **Fig. 4**

Added the thermal power with CCS and nuclear power uncertain analysis.

Fig. 4 Uncertainty analysis of the development of promising technologies. a1-f1 shows the kernel density estimation results for technology development. The black line in the graph represents

the first, intermediate range (IQR), and third quantile of all the cases. **a2-f2** show the development of the PV power installed capacity (unit: TW), wind power installed capacity (unit: TW), annual energy storage technology usage (unit: PWh), BECCS power installed capacity (unit: GW), thermal power with CCS installed capacity (unit: GW) and nuclear power installed capacity (unit: GW), respectively. The box plot shows the first, intermediate range (IQR), and third quantile of all the results, where the data range within 1.5 times the IQR is shown by whiskers. The thick blue line represents the pathway of the NDC scenario, and the thick grey line represents the pathway for the intermediate case of each scenario. The divergent colour from blue to green reflects the increasing stringency in the cumulative carbon budget. The cumulative CO₂ parameter corresponds to the absolute value of China's cumulative carbon budget for 2010-2050. **a3-f3** represents the relationship of the development of each technology with input uncertainty parameters, as fitted by a linear function. Three parameters with significant impact among fourteen uncertain input parameters are shown for each technology. The remaining results are shown in Supplementary Figs. 5-10. The variables (except for the cumulative carbon budget) represent multiples of the intermediate cases for each scenario. The intermediate case of PEAK20, PEAK25, and PEAK30 are devoted by "□", "+", and "×", respectively.

Supplementary Fig. 9

Added the relationship plot of thermal power with CCS and uncertain variables.

Supplementary Fig. 9 Scatter plot and linear regression results between thermal power with CCS capacity and Latin hypercube sampling uncertain variables. The intermediate cases of PEAK20, PEAK25, and PEAK30 are represented by "□", "+", and "×".

- **Supplementary Fig. 10**

Added the relationship plot of nuclear and uncertain variables.

Supplementary Fig. 10 Scatter plot and linear regression results between nuclear power capacity and Latin hypercube sampling uncertain variables. The intermediate cases of PEAK20, PEAK25, and PEAK30 are represented by "□", "+", and "×".

● **Supplementary Information, Page 3**

Added the statistical summary of the newly added uncertain input parameters.

Supplementary Table 2 Statistical summary of uncertain input parameters

Parameters	Distribution	Min	1-quartile	Mid	3-quartile	Max	Mean	SD
Bio Cap	Log-normal	0.700	0.926	1.000	1.080	1.430	1.007	0.117
BECCS Cost	Log-normal	0.700	0.925	1.000	1.080	1.430	1.007	0.117
Solar Cap	Normal	0.700	0.935	1.000	1.070	1.300	1.000	0.097
Solar Cost	Log-normal	0.700	0.926	1.000	1.080	1.430	1.007	0.117
Storage Cost	Log-normal	0.700	0.926	1.000	1.080	1.430	1.007	0.117
Wind Cap	Normal	0.800	0.956	1.000	1.040	1.200	1.000	0.065
Wind Cost	Log-normal	0.800	0.953	1.000	1.050	1.250	1.003	0.072
Thermal CCS Cost	Log-normal	0.700	0.926	1.000	1.080	1.430	1.007	0.117
Hydrogen Cost	Log-normal	0.700	0.926	1.000	1.080	1.430	1.007	0.117
Industry CCS Cost	Log-normal	0.700	0.926	1.000	1.080	1.430	1.007	0.117
Nuclear Cap	Normal	0.700	0.935	1.000	1.070	1.300	1.000	0.097
Nuclear Cost	Log-normal	0.700	0.926	1.000	1.080	1.430	1.007	0.117
Price Elasticity	Normal	0.700	0.935	1.000	1.070	1.300	1.000	0.097
Carbon Budget	Uniform	240.0	255.0	270.0	285.0	300.0	270.0	17.34

Comment 1.3:

I understand China-TIMES is a massive model, and this paper focus on MCA, however, it would be good the paper can add a session to introduce more about the model, and direct the reader material/manual/paper that has detailed documentation of the formula, parameters, and sources of

the data, etc. What's the base year? What discount does the model use? In addition, what software, environment, and resources are needed to run China-TIMES-MCA, will the model and data be open-source? Those could help the readers understand better of the model.

Response:

Thank you for your valuable suggestion. Some literature (from 17 to 26) cited in this paper are our studies using the China-TIMES and China MARKAL models for different sectors and concerns. But this may have some inconspicuous, so we have added a subsection dedicated to the China-TIMES model in the methodology section, and we will also include links to more detailed TIMES documentation for further understanding.

The base year for the China-TIMES model is 2015. The model uses 5% as the discount rate. The TIMES source code is open source and is available free of charge under a GPL v3.0 license. However, the user interface (VEDA/ANSWER) and solver (GAMS/CPLEX) are licensed for use.

As for China-TIMES-MCA, the Latin Hypercube Sampling (LHS) used in our paper is publicly available from Sandia National Laboratory. Nowadays, Python and R both have libraries to perform LHS. For the model data transfer and input file generation, we used MATLAB, which also can be implemented in other popular programming environments. For the organization and visualization of the results, we used R, Visio, Tableau Desktop, and Tableau Prep, which are licensed for free for students at accredited academic institutions. It took 6-8 hours to complete all 3,000 scenario simulations using the CPLEX solver on a PC (Intel i7-9700, 32GB RAM, 1T SSD). The time may vary with different solvers and different computer performance.

The model is calibrated with China Energy Statistics Yearbook. The data for the key uncertain variables discussed in the paper are disclosed in the Supplementary Information. Additional data that support the findings of this study are available from the corresponding author upon reasonable request.

Changes/additions to the manuscript:

● **Manuscript, Page 22, Line 399-400**

The base year for the China-TIMES model is 2015. The model uses 5% as the discount rate.

● **Manuscript, Page 26, Line 479-483**

On a PC with Intel i7-9700, 32GB RAM, 1T SSD, a round of optimization using CPLEX takes 6-8 hours, and the performance of the computer and solver can have a significant impact on the computing time.

● **Manuscript, Page 26, Line 484-493**

Data availability

The energy system part of China-TIMES is calibrated with China Energy Statistics Yearbook, which is a public publication. Local air pollutants are calibrated according to the "Second National Pollution Source Census Bulletin", which can be viewed on the website of the Ministry of

Ecology and Environment

(http://www.mee.gov.cn/xxgk2018/xxgk/xxgk01/202006/t20200610_783547.html). The data for the key uncertain parameters discussed in the text are disclosed in the Supplementary Information. Additional data that support the findings of this study are available from the corresponding author upon reasonable request.

● **Manuscript, Page 27, Line 495-502**

Code availability

The code and documentation of TIMES model is open source and is available on GitHub (https://github.com/etsap-TIMES/TIMES_model). The generator of Latin Hypercube Sampling used for generating uncertain input parameters publicly available from Sandia National Laboratory. For the model data transfer, input file generation and part of visualization, we used MATLAB and R, the code can be found on GitHub (<https://github.com/ZhangShuTHU/China-TIMES-MCA>). For the rest of the visualization, we used Tableau and Visio, with no programming involved.

New References in Manuscript

50. IEAGHG. *Towards Zero Emissions CCS In Power Plants Using Higher Capture Rates Of Biomass*. (IEAGHG, Cheltenham, 2019).
51. IEAGHG. *CCS In Energy And Climate Scenarios*. (IEAGHG, Cheltenham, 2019).
52. Zhang, P. *Evaluation of the Techno-Economics of Nuclear Hydrogen Production using HTGR (China)*. Report No. 1011-4289978-92-0-109318-9, 100-140 (International Atomic Energy Agency, Vienna, 2018).

● **Supplementary Information, Page 5**

Added data and sources for power sector investment projections.

Supplementary Table 3 Technology investment cost projection of the intermediate case in the China-TIMES-MCA model

Investment cost (USD/ kW)		2020	2025	2030	2035	2040	2045	2050
Thermal power without CCS ^{1,2}	Biomass & coal co-combustion	716	698	681	664	664	664	664
	Ultra-supercritical coal-fired power	550	547	545	542	540	538	535
	IGCC coal-fired power	1,454	1,324	1,206	1,098	1,000	911	829
	Nature gas steam turbine power	550	550	550	550	550	550	550
Thermal power with CCS ^{1,2,3,4}	Biomass & coal co-combustion w/ CCS retrofit	1,021	947	878	835	794	774	755
	Biomass combustion w/ CCS	2,128	1,916	1,726	1,554	1,400	1,261	1,136
	Ultra-supercritical coal-fired power w/ CCS	1,062	1,031	1,000	970	941	913	886
	Oxygen enriched coal-fired power w/ CCS	1,062	1,039	1,017	995	974	953	933
	IGCC coal-fired power w/ CCS	2,107	1,868	1,655	1,467	1,300	1,152	1,021
	NGCC gas-fired power w/ CCS	921	912	904	897	897	897	897
Nuclear ^{2,5,6}	Pressurized water nuclear power	1,995	1,946	1,897	1,850	1,805	1,805	1,805
	High temperature gas cooled nuclear power	2,857	2,571	2,286	2,171	2,057	1,943	1,829
Renewables ^{2,5}	PV power	754	637	538	454	383	324	324
	Offshore wind power	1,980	1,859	1,746	1,640	1,540	1,446	1,358
	Onshore wind power	1,095	1,056	1,019	983	948	915	882
	Small hydro power	1,312	1,312	1,312	1,312	1,312	1,312	1,312

	Large hydro power	1,066	1,066	1,066	1,066	1,066	1,066	1,066
Energy storage ^{2,8}	Compressed air energy storage	824	741	666	629	594	560	529
	Flow cell battery energy storage	1,064	818	782	668	570	486	415
	Flywheel energy storage	273	238	208	190	174	159	146
	Lithium battery energy storage	5,235	4,352	2,712	2,271	1,892	1,577	1,293
	Lead battery energy storage	1,198	1,072	725	662	599	536	505
	Pumped hydro storage	885	821	762	742	723	704	686
Hydrogen ^{1,8}	Water electrolysis	1,138	837	691	539	420	327	255

Note: the data sources are as follows.

1. IEA. *Energy Technology Perspectives 2020*. (IEA, Paris, 2020).
2. CEC. *China Power Industry Annual Development Report 2020*. (China Building Materials Press, Beijing, 2020).
3. IEAGHG. *Towards Zero Emissions CCS In Power Plants Using Higher Capture Rates Of Biomass*. (IEAGHG, Cheltenham, 2019).
4. IEAGHG. *CCS In Energy And Climate Scenarios*. (IEAGHG, Cheltenham, 2019).
5. IEA. *World Energy Outlook 2020*. (IEA, Paris, 2020).
6. Zhang, P. *Evaluation of the Techno-Economics of Nuclear Hydrogen Production using HTGR (China)*. Report No. 1011-4289978-92-0-109318-9, 100-140 (International Atomic Energy Agency, Vienna, 2018).
7. IRENA. *Electricity Storage Valuation Framework: Assessing System Value and Ensuring Project Viability*. (International Renewable Energy Agency, Abu Dhabi, 2020).
8. IEA. *Projected Costs of Generating Electricity 2020*. (IEA, Paris, 2020).

Comment 1.4:

Name in Supplementary Fig. 1 China-TIMES-MCA structure: I suggest using “4 Deterministic Scenarios” and “3000 Uncertain Cases”, so to differentiate the different focuses. I’m not clear about how the 3000 cases are generated? How the 8 key parameters’ probability distribution change over the different cases? Random? Simultaneous? Combination?

Response:

Thanks for your comments on this paper improvement. We have revised the statements in the Supplementary Figure 1. To give the reader a more intuitive idea of the framework of our model, we have moved the figure from the supplementary information to the main text (Figure 1).

The settings of the different key uncertain parameters are shown in Supplementary Table 2. Specifically, according to the nature of different parameters, we set up uniform, lognormal and normal distribution, while ensuring that the median of the parameter is 1.

Take the PEAK20 scenario as an example. We draw 1000 samples in a fourteen-dimensional vector space (the key uncertain parameters). The steps of Latin Hypercube Sampling (LHS) are: (1) divide each dimension into 1000 intervals that do not overlap each other, such that each interval has the same probability; (2) randomly draw a point in each interval in each dimension; (3) then randomly draw the points selected in (2) from each dimension and compose them into a vector. Therefore,

different parameters are randomly generated in the same scenario. And the same case in different scenarios, they are exactly the same (such as the 1st case of PEAK20 and the 1st case of PEAK25), so as to better compare the differences of scenarios.

Changes/additions to the manuscript:

- **Manuscript, Page 23-24, Line 432-438**

According to the nature of different parameters, we set up uniform distribution, lognormal distribution, and normal distribution, while ensuring that the median of the distribution is 1. Different parameters are randomly generated in the same scenario. The sample probability distributions and statistics summary are shown in Supplementary Table 2. The parameters of the case with the same order number are identical for different scenarios (such as the 1st case of PEAK20 and the 1st case of PEAK25), so as to better compare the differences among scenarios.

- **Fig.1 (the original Supplementary Fig. 1)**

Moved Supplementary Fig. 1 to main text as Fig. 1. The figure has been modified according to the response of Comment 1.2.

Fig. 1 China-TIMES-MCA structure. This figure shows the necessary information about the China-TIMES-MCA model. The dark blue part is the uncertain case generator and energy system optimizer. The blue part shows the parameters obtained from the statistical data and literature for the intermediate case. The brown part illustrates the scenario design of the study. The turquoise part is the end-use demand considering price elasticity. The red part is the fourteen input parameters with the probability distribution. The green part is the model result, which corresponds to our main findings in this paper. Finally, the model gets four deterministic scenarios and 3,000 uncertain cases for the analysis.

Comment 1.5:

What emission factors does the paper use when calculating emissions? Assuming the same emission factors over time? Or different ones?

Response:

Thank you for your comments. For CO₂, we used the emission factors published by the IPCC. For local air pollutants, the emission factors of different fuels in different sectors were mainly referred to the settings of GAINS models, and the models were calibrated with reference to the Second National Pollution Source Census Bulletin.

All emission factors are assumed to be constant. The emission factor of the local pollutant is set to be constant because the effect of terminal treatment measures can be excluded, thus reflecting the synergistic effect of CO₂ mitigation actions.

Changes/additions to the manuscript:**● Manuscript, Page 23, Line 420-422**

Emission factors for CO₂ are set according to IPCC guidelines⁴¹ and remained constant over time. Emission factors for local air pollutants are calibrated with official statistics and kept constant over time to correctly reflect the synergistic effects of CO₂ emission reductions.

New References in Manuscript

41. IPCC. *Revised 1996 IPCC Guidelines for National Greenhouse Gas Inventories: The Workbook (Volume 2)*. (IPCC, Geneva, 1996).
-

Comment 1.6:

Given this MCA practice, I'm looking for some well-summarized conclusions that from the simulation. Which scenario do the authors promote? What are the key uncertainties/factors for policy making?

Response:

Thank you for providing us with directions to improve the paper. Our study follows a "what-if" approach, so that policy makers can know the consequences of different peak times and cumulative emissions targets for the energy transition and climate governance. The impact of uncertainty in promising technologies on pathways was also assessed.

In the revised version of the paper, we compare the differences in pathways at different times of peaking (different scenarios) in the *CO₂ emission reduction pathway* subsection, showing that peaking in near 2025 (PEAK25 scenario) is a good choice that combines short-term pressures and long-term goals. In the in-depth specific energy decarbonization analysis that follows, it is also shown that early peaking has the effect of reducing welfare losses, lowering the cost of carbon to society, reducing reliance on negative emissions, and increasing synergies.

We have carefully combed through the language of the manuscript and highlighted some distinctive points. The most important uncertainties are the timing of peaking and the cumulative carbon emissions target which profoundly affect the future development. Therefore, we have formed the first two policy insights to address this reality, i.e., designating transparent cumulative emission

targets to balance the near- and long-term action and enhancing NDC to increase near-term efforts.

In addition to this, the uncertainty of the technology is assessed. We find that the decreasing cost of hydrogen production and CCS technologies has a very significant effect on future marginal abatement costs, and welfare loss reduction. This becomes the reason for the third policy recommendation (advanced technology innovation) in the paper.

We have restructured the *Result* section and *Discussion and policy insight* section of the manuscript to better highlight the core claims of our paper. Listed below are the findings that we believe are consistent with the model result. In our responses to Comment 2.3, and 3.4, you can also find views on what we are promoting and what we suggest policymakers need to do.

Changes/additions to the manuscript:

- **Manuscript, Page 5, Line 68-69**

Peaking earlier and lower can undoubtedly alleviate the pressure of the subsequent transition, but also frames the great challenge of near-term mitigation actions.

- **Manuscript, Page 7, Line 111-113**

Since non-carbon emissions are considered difficult to completely remove in current perceptions, this requires more ambitious CO₂ reductions to achieve 1.5-degree and China's carbon neutrality target.

- **Manuscript, Page 8, Line 130-133**

In 2030, the installed capacity of wind and solar reaches 1.2 TW under the NDC scenario and about 2 TW under the PEAK20 and PEAK25 scenarios, suggesting that renewable energy development is not on track for carbon neutrality under the current NDC target.

- **Manuscript, Page 12, Line 203-211**

Through the analysis of a large number of cases, we can summarize the following robust findings. First, renewable energy should and will grow fastest among the energy supply technologies compared to the NDC scenario (without further policies). Second, after 2030, coal-fired power without CCS will rapidly diminish, and BECCS will gradually gain popularity after 2035 and become crucial by the middle of this century. Third, thermal power with CCS and energy storage technologies that can provide power system stability and reliability in the future will receive greater development attention. Fourth, with the exception of nuclear power, which has limited potential due to siting constraints, all other technology developments are greatly influenced by the ambition and commitment to climate action.

- **Manuscript, Page 18, Line 316-319**

Cost reduction of CCS, hydrogen energy, can significantly reduce the marginal abatement costs in 2050, which illustrates the importance of technological innovation for much-needed technologies to reduce the policy costs of climate governance.

● **Manuscript, Page 19, Line 350-352**

Achieving a carbon peak near 2025 with a peak around 10.3 GtCO₂ and declining to 8.2-8.7 GtCO₂ in 2030 is the possessive choice to combine near-term and long-term transition pressures.

Comment 1.7:

Fig 2. Change the order of technology in legends to the same order as in the stacked area chart so to make it more readable. The unit of power installed capacity is TW, not GW.

Response:

Thank you for helping us discover the clerical error. We have revised the legend and text to deliver the information more clearly to the reader.

Changes/additions to the manuscript:

● **Fig. 3 (the original Fig. 2)**

Modified the order of the legend.

Fig. 3 Graphic of power installed capacity (unit: TW), power generation (unit: PWh), primary energy mix (unit: EJ), and final energy mix (unit: EJ) for the intermediate case of each scenario. For the power installed capacity, the stacked area chart shows different power plant types, and the black line represents the proportion of the renewable energy contribution to the total capacity. For the power generation, the stacked area chart shows the annual power generation of different types of power generation technologies. The black line represents the proportion of renewable energy power generation to the total power generation. For the primary energy mix, the stacked area

chart represents different types of primary energy consumption, and the black line represents the renewable energy share. The calorific value calculation method is applied for energy statistics. For the final energy mix, the stacked area chart shows the end-use of different energy types, and the black line represents the electrification rate.

Comment 1.8:

Supplementary Table 1, are 2010, 2015, 2020 historical actual data or assumption data? It might be useful to note the differences if are assumption, and could be used to compare the modeling behavior and performance.

Response:

Thank you for arising this issue. The data for 2010, 2015 are the official data. The statistical data release in China consists of two steps: preliminary accounting and final verification. Since the National Bureau of Statistics (NBS) has not yet released the final verified data, the data for 2020 are assumed with reference to the NBS preliminary accounting data.

Changes/additions to the manuscript:

● **Supplementary Information, Page 2**

Note: The data in the table for 2010 and 2015 are the final validated data from the National Bureau of Statistics, the data for 2020 are the preliminary accounting statistics from the National Bureau of Statistics, and the data for 2025 and beyond are projections based on domestic experts.

Comment 1.9:

Supplementary Table 3. What are the cost assumptions of conventional technologies: coal, gas, nuclear, hydro technologies? Coal combustion with CCS cost seems too low? Could you please provide the sources/rationales of those numbers?

Response:

Thank you for your comments. The detailed cost assumptions for various power supplies are shown in the table below. We will also add data sources.

In our initial submission, we did not disclose cost data for thermal power with CCS. The *biomass & coal co-combustion with CCS* covered in Supplementary Table 3 represents the cost of retrofitting existing coal-fired thermal power to biomass & coal co-combustion plant with CCS, which we will further elaborate.

However, as the authors note, China's coal power investment cost is significantly lower than the world average, with a 1000MW ultra-supercritical unit, for example, costing about 3,300CNY/kW (~550USD/kW) in 2019, so even with the addition of CCS, its costs will likely be lower than the world average. To address this situation, we refer to the cost increase of coal-fired power with CCS relative to coal-fired power in the IEA report and make assumptions about future coal-fired power with CCS costs based on actual coal-fired power costs in China.

In the revised version, we include thermal power with CCS costs as a key uncertain variable as well. The results show that due to the need to achieve rapid emission reductions in China over the next 40 years, the development of thermal power with CCS is more limited for the vast majority of the cases even when the cost of thermal power with CCS is significantly lower than the world average, and renewable energy and BECCS are more promising for large-scale development.

Changes/additions to the manuscript:

- **Supplementary Information, Page 4**

We summarized investment cost data for the power sector and data sources placed in Supplementary Table 3. Reviewers can also find the revised table in our response to comment 1.3.

Reviewer #2:**General comment:**

This paper conducts a Monte Carlo analysis on several key parameters using the China-TIMES model. Several thousand scenarios are generated, and basic statistics are carried out on the scenario results, in order to identify important insights for the evolution of China's energy mix and emissions out to 2050.

Response:

We would like to thank the respected reviewer for giving us the opportunity to improve our work.

Comment 2.1:

The methodology employed in this study is solid - for the most part. The China-TIMES model is well-documented and well-respected in the literature. The novel methodological aspect of this study is the use of Monte Carlo analysis, which the authors perform in a straightforward way. The one issue I have with the study approach, however, is that the number of uncertain parameters that are varied in the Monte Carlo analysis are relatively few (8 in total), and pretty much all of them are focused on power sector technologies (Supplementary Table 2). It would be important to also include other uncertain parameters in the Monte Carlo analysis, such as the techno-economics of various end-use sector technologies, or perhaps the price elasticities on service demands.

Response:

Thanks for your suggestion. In the revised version of our paper, we added 6 new parameters to highlight the focus of energy transition. Among the six parameters, the cost of hydrogen (production, storage and transportation, distribution), the cost of industry CCS use and the price elasticity are included.

Through additional research, we find that the use of hydrogen emerges as a backstop energy source, mainly to address areas that are difficult to electrify in the end, rather than being promoted on a large scale, as some researchers believe. We have added a new figure (Fig. 6) to illustrate the production and consumption of hydrogen energy, and how it is affected by other uncertain parameters. More explanations can also be found in our response to Comment 3.3.

The higher the elasticity of demand in the model, the more it is affected by price changes. But a small reduction in demand can bring about a significant reduction in marginal abatement costs for some scenarios. This implies that for some of the industries that are difficult to reduce emissions, effectively guiding and managing demand and providing alternative service demand solutions are also important strategies to help carbon neutrality. We have added a discussion of the effects of price elasticity uncertainty in the paper, added a figure (Supplementary Fig. 14) to illustrate the impact of price elasticity on important energy and economic variables, while modifying Figure 5 (original Figure 4).

Changes/additions to the manuscript:

● **Manuscript, Page 13-14, Line 229-238**

The uncertainty in price elasticity has a greater impact on industry and transport sectors but a smaller impact on the building sector probably due to the longer life span of equipment in the building sectors. The closer we get to carbon neutrality, the greater the uncertainty of demand changes. On the one hand, this stems from the fact that stringent climate policies have pushed up the cost of energy services in high-emitting sectors, thus cutting demand due to the price rise, and on the other hand, as society evolves, there are more alternative options to meet energy service demands, thus increasing price elasticity significantly. More elastic demand, while reducing energy consumption and the cost of achieving deep emission reductions to a greater extent, entails a faster rise in welfare losses (Supplementary Fig. 14).

● **Fig. 5 (the original Fig. 4)**

Added the effect of price elasticity on demand.

Fig. 5 Graphic of the impact of mitigation on the producer and consumer behaviours for the intermediate case of each scenario. **a1-c1**, The decline in the rate of demand for the industry, building, and transport sectors due to price elasticity relative to the NDC scenario (unit: %). **a2-c2**, represents the relationship of the demand with input uncertain price elasticity. **d**, The total discounted welfare loss relative to the NDC scenario for 2020-2050 (unit: trillion US dollar). The black spot in panel **d** denotes the welfare loss for the intermediate case of each scenario. The box plot shows the first, intermediate range (IQR), and third quantile of all the results, where the data range within 1.5 times the IQR is shown by whiskers. The divergent colour from blue to green reflects the increasing stringency in the cumulative carbon budget. The cumulative CO₂ parameter corresponds to the absolute value of China's cumulative carbon budget for 2010-2050.

● **Supplementary Fig. 14**

Added the relationship plot of price elasticity and important energy and economic variables.

Supplementary Fig. 14 Scatterplot of primary energy supply, final energy consumption, marginal abatement cost, welfare loss and price elasticity. The shadows of different concentrations indicate the positions of ten quantiles of variables. The divergent colour from blue to green reflects the increasing stringency in the cumulative carbon budget. The cumulative CO₂

parameter corresponds to the absolute value of China's cumulative carbon budget for 2010-2050. Black dots represent the intermediate cases.

Comment 2.2:

The results of the paper are very important, not just for China but for the world at large. The paper reports a huge amount of results, both in text and figures. However, the paper is not told as a story, but is rather diagnostic in its presentation of results. This will lose the attention of many readers very quickly. It would be preferable to highlight the importance of certain results as they are described, perhaps mentioning when things are counter-intuitive and/or how they compare to other recent studies found in the literature (not just for China).

Response:

Thanks for your highly valuable suggestions, we have rewritten the *Result* section and the *Discussion and policy insight* sections and analyzed the profound meaning behind the numbers.

In the manuscript, our results (emission pathways, demand sector electrification rates, renewable energy penetration, and local air pollutant synergies) are compared with papers published in top journals containing either the China carbon neutrality scenario or the 1.5-degree scenario, with consensus among us, and the reliability of these assertions is further validated by the large number of uncertainty scenarios in this study.

In addition, as a result of our uncertainty analysis, we also identified such robust and interesting findings as 1) the urgency of reducing emissions from buildings in the near term is much greater than that of transportation; 2) the shift in the role of wind power from cooperation to competition with "PV + energy storage" in the process of renewable energy expansion; 3) the rapid decline in load factor that coal-fired power plants may face in 2035 under the carbon neutral scenario and 4) the incremental renewable energy investment required to peak in 2025 compared to 2030 is only one-fifth of the incremental future welfare loss due to late peaking.

Based on our findings, we offer policy insights such as clear cumulative emissions targets, enhanced near-term actions for early peak attainment, concerted efforts in the energy supply and demand sectors, and green investments to support policymakers in deploying future decarbonization strategies.

We have made extensive changes to the text of the full manuscript, and the following excerpts show some noteworthy findings from our comparison with other studies and our research.

Changes/additions to the manuscript:

● **Manuscript, Page 5, Line 64-68**

The cumulative emissions of the feasible cases (675 out of 1000) are all above 259 GtCO₂, suggesting that it will be difficult to reach the 1.5-degree goal without substantial negative emission technologies (NETs) in the second half of the century, which in line with the multimodel comparison result^{29,30}.

- **Manuscript, Page 6, Line 77-80**

These aggregate pathways, however, hide the dynamics of the emission reduction process among sectors. Take the emission peaking time as an example, power sector, the current largest emitter, has the greatest near-term mitigation potential, and the timing of its peak determines the timing of total CO₂ emissions peak.

- **Manuscript, Page 7, Line 96-99**

The building and transport sectors are in quite different status. The sluggish growth in building floor area and high electrification rate in the building sector, while the continuous growth of transport demand and less than 4% electrification rate in the transport sector, has resulted in very divergent development trends.

- **Manuscript, Page 9-10, Line 152-157**

Wind power and "PV power + energy storage" have a clear complementary relationship, jointly replacing fossil fuel before 2040. After 2040, wind power competes with "PV power + energy storage" for the nighttime load supply (Supplementary Figs. 1 and 2). We also note that hydrogen production and electric vehicle (EV) charging might affect the load characteristics, such that load management can significantly reduce the demand for energy storage.

- **Manuscript, Page 10, Line 160-163**

Significant heterogeneity between power generation and the installed capacity reflects the rapidly declining capacity factor of coal-fired power, from 0.4 in 2025, to 0.3 in 2030 and 0.1 in 2035, further exacerbating the urgent issue of thermal power retirement and transition.

- **Manuscript, Page 14, Line 247-249**

The large gap between the present and the future reflects the need for electrification to be accelerated and enhanced in the future, which is all agreed upon in the literature³⁶.

- **Manuscript, Page 16, Line 280-282**

Understanding that the energy transition creates a huge need for investment and technological innovation that drives economic growth, but also brings economic burden and welfare loss.

- **Manuscript, Page 16-17, Line 291-295**

Considering the whole energy system, on average, the cumulative GDP loss for 2020-2050 due to energy system investment, maintenance and operation costs in the mitigation scenarios is 3.3%-3.6%, compared to 3.2% for NDC scenario. Significant reductions in operating costs offset most of the rising investment costs. The small increase in total energy system costs also shows that China's energy transition is achievable.

- **Manuscript, Page 18, Line 324-326**

And in 2050, local air pollutant emissions for almost all scenarios fall to only one-fifth of current values (Fig. 9). The results are very consistent with the previous study when carbon neutrality is achieved³⁷.

New References in Manuscript

29. Duan, H. *et al.* Assessing China's efforts to pursue the 1.5 degrees C warming limit. *Science* **372**, 378-385 (2021).
 30. van Soest, H. L., den Elzen, M. G. J. & van Vuuren, D. P. Net-zero emission targets for major emitting countries consistent with the Paris Agreement. *Nat. Commun.* **12**, 2140 (2021).
 37. Cheng, J. *et al.* Pathways of China's PM2.5 air quality 2015–2060 in the context of carbon neutrality. *Natl. Sci. Rev.*, nwab078 (2021).
-

Comment 2.3:

Moreover, the paper is missing some punch lines. What are the main messages that the reader should take away? Which uncertainties matter more than others? Could the authors even do something more sophisticated in their analysis of scenario results – something like Robust Decision Making / Scenario Discover (Lempert et al. papers) – in order to identify the most impactful uncertainties? I would like to highlight that the insight on p. 12 (also in the abstract) about nearer-term peaking resulting in lower welfare loss and less reliance on carbon dioxide removal is a crucially important finding.

Response:

Thank you for your suggestions.

We added six new key uncertain parameters to the model. Out of a total of 14 parameters, we find that the uncertainty in cumulative emissions has the largest effect on the pathway. Given the strong link between cumulative GHG emissions and global warming, and the fact that China has not yet set targets on cumulative or annual emissions, transparent and detailed long-term targets are an important means of addressing uncertainty.

For technology-related uncertain parameters, we find that for all technologies, the increased economic potential reduces the transition challenges, but the cost reduction is more significant for hydrogen and CCS technologies. As for nuclear energy, it is mainly constrained by the choice of site (mainly by policy and public acceptance), and has little to do with its own cost reduction.

The timing of peak attainment has a significant impact on the long-term benefits and losses of climate action. A later peak, with a defined cumulative emissions constraint, is more likely to achieve a policy goal like carbon neutrality, but does not have as strong an effect on climate change mitigation as an earlier peak. Thus, there is a trade-off between faster achievement of policy goals and greater mitigation contributions.

The following are other consistent messages we obtain when all these uncertainties are taken into account, which we hope will be useful for policy makers.

First, a later peak would be more dependent on negative emission technologies, but achieve carbon neutrality sooner. Second, regardless of when overall emissions peak, the power system needs to be

net-zero between 2040-2045, and the later we act, the less time we have left to transition the power system. Third, our research shows that CCS technology is indispensable in the future, so it is necessary to deploy CCS pipeline network construction and industrial pilots in advance.

In our revised manuscript, we have summarized these important findings and some noteworthy issues. In our responses to Comment 1.6 and 3.4, you can also find our modifications to the policy insights.

Changes/additions to the manuscript:

- **Manuscript, Page 5-6, Line 74-76**

A paradox can be seen in the trade-off between early mitigation actions (greater mitigation contribution) and long-term dependence on NETs (earlier achievement of carbon neutrality).

- **Manuscript, Page 6, Line 85-89**

Despite the emission peaking time of power sector varies considerably, one highly consistent finding is that power sector emissions turn negative typically in 2040-2045, with emissions of -1.3 to 0.1, -1.6 to 0.2, -1.6 to -0.5 GtCO₂ in 2050 for PEAK20, PEAK25, and PEAK30 scenarios. This inspires the need to decarbonize the power sector in a timely manner. The later we act, the less time we have left to transition the power system.

- **Manuscript, Page 7, Line 104-106**

This foreshadows that the building sector need to act faster than the transport sector in the near term, but transport sector emission reductions require full attention in the long term.

- **Manuscript, Page 10, Line 163-168**

The cautionary tale is that coal-fired power is rebounding at a time when coal power needs to be controlled for development. Compared to pre-pandemic levels, coal power generation growth in China outpaced wind and solar in the first half of 2021 and China's energy supply sector did not "green recovery" as hope³². With 292 GW of new coal-fired power plants is currently announced, permitted, shelved, and under construction in China³², stakeholders do need to reassess the long-term risks involved.

- **Manuscript, Page 15-16, Line 274-277**

Since CCS technologies require extensive construction of the pipeline and the infrastructure, given the great expectations for CCS technology in both the energy supply and industry sectors, the layout of the pipeline network and the commercial promotion are essential for the CCS popularization.

- **Manuscript, Page 19, Line 347-350**

Recently, major economies such as the US, EU, Japan and Korea have updated their NDCs ahead of the COP26 meeting. For China, an enhanced NDC would boost climate mitigation while facilitating the domestic transition. It is wise choice for China to enhance NDC, aiming to reach the emission peak earlier and lower.

- **Manuscript, Page 20, Line 358-361**

For China, an appropriate cumulative emission target should dovetail with the announced carbon neutrality target in a way that does not impose an excessive burden on society, but also appropriately bears international responsibility and contributes to the 1.5-degree target.

New References in Manuscript

32. Dave, J., Nicolas, F. & Peter, T. Global Electricity Review: H1-2021. (Ember, London, 2021).
-

Comment 2.4:

The quality of written English in the paper is very good, and the presentation and organization work very well. Some more specific comments below ...

Response:

We appreciate the reviewer's comments. We have responded to the subsequent comments mentioned and elaborated them one by one.

Comment 2.5:

Page 4 => It would be helpful to have a table (or diagram) here that clearly explains the scenario design.

Response:

Thank you for your comment. We originally placed the model framework diagram in the Supplementary Information. Now, we have placed Supplementary Fig. 1 in the main body to give the reader a clearer understanding of the model's framework and scenario design.

Changes/additions to the manuscript:

- **Fig. 1 (the original Supplementary Fig. 1)**

Moved the original Supplementary Fig. 1 to the Fig. 1 in the main body. The other figures are reordered.

Comment 2.6:

Page 5, first paragraph => These are fossil fuel and industrial sector emissions only, right? No land use emissions included here? It would be helpful to state this explicitly at the start, so that there is no confusion on the part of the reader.

Response:

Thank you for arising this issue. As stated by the reviewer, our study focuses on energy-related and process-based CO₂ emissions in China (including industry, buildings, transportation and energy supply sectors). The paper does not address LULUCF emissions or sinks. We state this explicitly in the CO₂ emission reduction pathway part.

Changes/additions to the manuscript:

● **Manuscript, Page 5, Line 57-58**

CO₂ emissions are the bulk of China's greenhouse gas (GHG) emissions and the focus of this study.

● **Manuscript, Page 6-7, Line 91-96**

Through the development of alternative materials, popularization of carbon capture and storage (CCS) technology and declining demand for energy-intensive products, as seen from the result for 2050, energy-related emissions from industry sector will drop to 0.4-1.7 (PEAK20), 0.3-1.5 (PEAK25), and 0.3-1.1 GtCO₂ (PEAK30) and industrial process emissions will reduce by 72-91% compared to 2020, to as low as 0.1 GtCO₂.

● **Fig. 2 (the original Fig. 1) title**

Sectoral energy-related CO₂ emission pathways under broad cumulative carbon budget range under different scenarios (unit: GtCO₂).

Comment 2.7:

Page 5, second paragraph => Please state clearly whether the 240-300 GtCO₂ carbon budget is for China or for the world as a whole.

Response:

Thanks for your comment. The 240-300Gt figure does tend to cause uprisings. All statements about carbon emissions, cumulative emissions and carbon budget in the text are specific to China. We have added descriptions where the cumulative carbon budget is involved.

Changes/additions to the manuscript:

● **Manuscript, Page 5, Line 59-61**

If existing NDC target is tracked, emissions will peak at 10.4 GtCO₂ in 2030 and then decline steadily to 7.3 GtCO₂ in 2050 with China's cumulative emissions of 381.1 GtCO₂ for 2010-2050.

Comment 2.8:

Page 5, second paragraph => There is a comma (,) before the word 'Note' . I think it should instead be a period (.).

Response:

Thank you very much for your comment. We have carefully checked for punctuation and spelling errors.

Changes/additions to the manuscript:

● **Manuscript, Page 5, Line 68-69**

Peaking earlier and lower can undoubtedly alleviate the pressure of the subsequent transition,

but also frames the great challenge of near-term mitigation actions.

Comment 2.9:

Page 10, middle of page => According to my understanding, these more fundamental changes in consumer behavior and lifestyle (remote office work, online shopping, sharing economy) are outside the scope of the price-induced demand responses modeled (i.e., the price elasticities do not cover those more major changes in demand). Some further explanation is needed on this point, in order for the reader to understand.

Response:

Thank you for your kind suggestions. The decline in demand in the transportation sector relative to the NDC scenario stems, on the one hand, from the increased energy costs of fuel car use (an aspect that can be explained by price elasticity) and, on the other hand, as suggested by the reviewers, from changes in people's lifestyles and consumption behavior. We have rewritten the description of the declining demand in the transportation sector in the manuscript.

Changes/additions to the manuscript:

- **Manuscript, Page 13, Line 228-229**

The turnover for light-duty vehicles in 2050 could fall by up to 9% compared to the NDC.

- **Manuscript, Page 14, Line 238-241**

Surely, the transition caused by price changes is far from being comparable to the changes caused by social changes, and can have a significant social cost. Profound changes in lifestyles and consumption concepts due to remote working, information connectivity, and the sharing economy are the cure for decarbonizing energy consumption.

Comment 2.10:

Page 11, middle of page => Are the investment numbers cumulative, undiscounted values? Over which timeframe (2010-2050)?

Response:

Thank you very much for raising this issue. This figure for the investment in the energy supply sector is a non-discounted cumulative investment for the years 2020 to 2050. We clarified this issue in the text.

Changes/additions to the manuscript:

- **Manuscript, Page 9, Line 148-150**

Fig. 4c show that energy storage usage rapidly increases until 2045, followed by a slowdown in growth to approximately 15% of the total generation in 2050, requiring 183-220 billion US dollars in investment during 2020-2050.

- **Manuscript, Page 16, Line 282-284**

The aggregate results show that the energy supply sector requires an investment of 4.9~7.8 trillion US dollars during 2020-2050 to kick-start the zero-carbon transition, representing an increase of at least 65% over that of the NDC scenario (Fig. 7).

Comment 2.11:

Page 17, top of page => Please be specific that the carbon budget is in terms of Gtons CO₂ from fossil fuel and industrial emissions (excluding land-use emissions), if I am correct.

Response:

Thank you for your comment. The carbon budget only includes CO₂ from fossil fuel combustion and does not include land use emissions. We specified the emissions where it relates to carbon budgets.

Changes/additions to the manuscript:

- **Manuscript, Page 25, Line 452-453**

The **carbon budget** parameter considers only energy-related CO₂ emissions and does not include LULUCF emissions.

Comment 2.12:

Page 17, bottom of page => I note that demand reduction is possible via price elasticities; however, vastly different demand trajectories have not been created (e.g., following different SSPs). The authors should state this explicitly, referring to Supplementary Table 1, either here or on p. 15.

Response:

Thanks for your valuable suggestions. We add references there for the value of elasticities and also include elasticities in the key uncertain parameters to better illustrate the impact of the magnitude of production and consumption pattern shifts on the transition. The results of the elastic uncertainty can be found in the response to comment 2.1.

Changes/additions to the manuscript:

- **Manuscript, Page 26, Line 477-478**

The choice of price elasticity of the intermediate case is based on the results of TIAM⁶⁰.

New References in Manuscript

60. Kesicki, F. & Anandarajah, G. The role of energy-service demand reduction in global climate change mitigation: Combining energy modelling and decomposition analysis. *Energy Policy* **39**, 7224-7233 (2011).
-
-

Reviewer #3

General comment:

This paper uses numerous scenarios generated by Monte Carlo method for several variables with pre-defined distributions. It is well written, but I have some significant concerns on the analysis in the paper.

Response:

Thank you for giving us the opportunity to improve our paper. We have carefully addressed your comments and hope that the revised paper is satisfying.

Comment 3.1:

First, the time horizon used for the analysis of the paper is 2010-2050, while China's carbon neutrality target year is 2060. In this sense, it will be much better to include both peak year and neutrality year in scenario design to deliver good insights on China's transition pathways.

Response:

Thanks for arising this issue. There are two main considerations for setting the time horizon at 2010-2050.

On the one hand, the time horizon of the current literature on cumulative carbon budget allocation is basically 2010-2050 or 2010-2100^{1,2}. The cumulative carbon budget is a very important uncertain parameter in this paper. To ensure the confidence of the parameter settings, we have limited the time horizon.

On the other hand, China's carbon neutrality target is for GHG emissions, while our research focuses on energy-related CO₂. China's special envoy for climate change Xie Zhenhua mentioned in a public speech in July that China's 2060 carbon neutral goal included GHG emissions from all sectors of the economy not only CO₂. Typically, net-zero emissions of energy-related CO₂ precede net-zero emissions of GHG. As our study mainly pays attention to energy system transition, achieving near zero or net zero energy-related CO₂ emissions by 2050 was the original intent of our scenario design.

After receiving your suggestion, we have integrated the results of other modeling groups in China³ and abroad⁴ on land use and non-CO₂ GHG emissions and added a discussion in the paper to remedy the shortcomings arising from the boundary setting of this study. The results show that China may achieve carbon neutrality by 2060 with the assumption that process-based CO₂ emissions and non-CO₂ GHG emissions can be offset by negative emissions from LULUCF (Land use, land use change and forestry)⁵. Therefore, the emission pathways of this study meet the requirement of China's carbon neutrality with a high probability.

References

1. Masson-Delmotte V, et al. *Global Warming of 1.5 OC: An IPCC Special Report on the Impacts of Global Warming of 1.5° C Above Pre-industrial Levels and Related Global Greenhouse Gas*

Emission Pathways, in the Context of Strengthening the Global Response to the Threat of Climate Change, Sustainable Development, and Efforts to Eradicate Poverty. (World Meteorological Organization, Geneva, 2018).

2. Robiou du Pont Y, Jeffery ML, Gütschow J, Rogelj J, Christoff P, Meinshausen M. Equitable mitigation to achieve the Paris Agreement goals. *Nat. Clim. Change* **7**, 38-43 (2016).
3. Institute of Climate Change and Sustainable Development Tsinghua University. *China's Long-term Low-carbon Development Strategies and Pathways Comprehensive Report.* (China Environment Publishing Group, Beijing, 2021).
4. Cheng J, *et al.* Pathways of China's PM2.5 air quality 2015–2060 in the context of carbon neutrality. *Natl. Sci. Rev.*, nwab078 (2021).
5. Wang J, *et al.* Large Chinese land carbon sink estimated from atmospheric carbon dioxide data. *Nature* **586**, 720-723 (2020).

Changes/additions to the manuscript:

- **Manuscript, Page 7, Line 107-113**

For China's carbon neutrality goal of net-zero GHG emissions from all economic sectors by 2060, a significant reduction in non-CO₂ GHGs is imperative. Given the extreme uncertainty of LULUCF (land use, land-use change and forestry) emissions, China's CH₄, N₂O and F-gas emissions excluding LULUCF, are 2.4 GtCO₂e in 2020, and at least 50% of the reductions needed by 2050 to meet stringent climate targets³¹. Since non-carbon emissions are considered difficult to completely remove in current perceptions, this requires more ambitious CO₂ reductions to 1.5-degree and China's carbon neutrality target.

New References in Manuscript

31. Institute of Climate Change and Sustainable Development Tsinghua University. *China's Long-term Low-carbon Development Strategies and Pathways Comprehensive Report.* (China Environment Publishing Group, Beijing, 2021).
-

Comment 3.2:

Second, I do not see obvious advantages of Monte Carlo method in such a type of analysis than well-designed much less sensitivity analysis. The authors do not explain in the SI on why and how they setup the distributions and relevant values for the 8 so-called key parameters. This is very crucial for the entire analysis and should be well described and explained.

Response:

We feel highly appreciated for your valuable questions as they effectively helped us introspect our shortcomings in explaining the distributions of key uncertain parameters which may prevent readers from understanding our result quickly and accurately.

As for parameter distribution selection, the distribution type is chosen to conform to the variation pattern of this parameter on the one hand, and to better illustrate the main idea of the article on the other hand. Among the 14 key uncertain parameters mentioned in the paper, they can be broadly classified into several categories. 1) Cost category. For the parameters related to the cost, we used a

log-normal distribution to reflect the percentage change relative to the median. 2) Resource category. The variables in this category reflect the resource endowment characteristics, and we chose a normal distribution to represent the error in the detected resource reserves. While the total resource of biomass is more related to the cost of the biomass recycling chain, combined with the assessment results for the global total biomass¹, we chose the log-normal distribution. 3) Price elasticity. Reflecting the dispersion of the price elasticity, the normal distribution is chosen. 4) Cumulative carbon budget. Since there is no universally accepted allocation scheme, we chose to include cumulative carbon budgets that would achieve the 1.5- and 2-degree targets, and therefore used a uniform distribution. In the figures we used color bands to characterize this parameter, illustrating the differences in results from different carbon budgets.

As for the LHS sampling, we used open-source software. For the program input and output structure, the program settings can be found in the documentation. We have added for distribution selection and establishment in the Supplementary Notes 1.

References

1. Hanssen SV, Daioglou V, Steinmann ZJN, Doelman JC, Van Vuuren DP, Huijbregts MAJ. The climate change mitigation potential of bioenergy with carbon capture and storage. *Nat. Clim. Change* **10**, 1023-+ (2020).

Changes/additions to the manuscript:

- **Supplementary Information, Page 4**

Supplementary Notes 1

The assumptions of the distributions follow the actual variation pattern of the parameters on the one hand, and to highlight the results of this study on the other hand. In the paper, a log-normal distribution is used for the cost-related parameters, which expresses in economic terms the effect of percentage changes in costs. Since there is no universally accepted national carbon budget allocation results, we chose a wide range of cumulative carbon budgets and used a uniform distribution to represent the effect of different cumulative carbon budgets on the pathways. For other parameters, we choose normal distribution to reflect the parameter uncertainty in a balanced way. When performing uncertain case generation, we fix the median (except Carbon Budget) to 1 and use it as a basis for other scenarios.

The uncertain case generator used in our study was developed by Sandia National Laboratories for the generation of multi variate samples by a constrained randomization termed Latin hypercube sampling (LHS). The generation of these samples is based on user-specified parameters which dictate the characteristics of the generated samples, such as type of sample (LHS or random), sample size, number of samples desired, correlation structure on input variables, and type of distribution specified on each variable. The following distributions are built into the program: normal, lognormal, uniform, log-uniform, triangular, and beta. In addition, the samples from the uniform and log-uniform distributions may be modified by changing the frequency of the sampling within subintervals, and a subroutine which can be modified by the user to generate samples from other distributions (including empirical data) is provided. The actual sampled values are used to form vectors of variables commonly used as input computer models for purposes of sensitivity and uncertainty analysis studies. The software code, documentation and input data can be found on

Comment 3.3:

Third, I do not think solar and wind potential are that important to be included in the 8 selected key parameters. The ultra-high voltage (UHV) technologies currently widely used in China could transmit large-volume remote PV and wind power in the north and west of China to demand centers in China's east and south, and the UHV network is planned to be largely expanded in the future. In this sense, as the model used by the authors has no sub-national resolution, I do not see any significant reasons to heavily analyze the solar and PV potential here. Moreover, I think that hydrogen use should be a key technology for deep decarbonization and for achieving carbon neutrality from carbon peak within a very short time period (like 30-40 years) in China, but it is not included in the currently designed Monte Carlo simulation.

Response:

Thank you for your valuable comments.

First, the reviewer expressed great interest in hydrogen, and we share this belief. Therefore, in our revised version of the paper, we took the advice and added the cost of hydrogen energy as a new key uncertain parameter to the model. The result shows that the large uncertainty range of hydrogen production. More detail information can be found in the revised manuscript and the following statements.

Second, we agree that there will be a lot of UHV construction and deployment in China in the future, which can solve a lot of energy spatial imbalance problems. As the consensus of academia and industry has reached, PV and wind power are the most promising power sources under the carbon neutral vision. The high penetration of renewable energy poses a challenge to power system reliability. Both in terms of operational security and capacity adequacy, there are drawbacks to variable renewables. Therefore, for the sake of power system security, the expansion of variable renewables needs to take into account the renewable energy accommodation and the heterogeneity of power generation periods (PV generates power during the day only, wind power generates more power at night, etc.). Hence, though China has huge reserves of wind and PV resources, the economically exploitable size is actually limited. In this paper, we consider the construction and operation of PV, wind and energy storage in the unified framework to obtain more realizable results for renewable energy and energy storage development. As the uncertainty of wind and PV capacity can greatly affect the construction and operation of energy storage, we believe that the potential of wind and solar is still valuable to discuss. This term "potential" may be misleading to the reader and will be further clarified in the revised version of the paper.

Changes/additions to the manuscript:

● **Manuscript, Page 14-15, Line 249-270**

Despite many studies suggest hydrogen is an indispensable option for decarbonization in a short period of time, different studies have not yet reached a consistent conclusion on the size of hydrogen energy in the demand sectors, mainly due to its high production costs and high

infrastructure investments. Fig. 6 reports our uncertainty analysis on hydrogen energy. First, we find climate goals are the most important factor influencing the expansion of hydrogen energy. Specifically, the need of hydrogen energy shows a significant exponential relationship with CO₂ emissions, and a decrease in the cost of hydrogen energy can expand the demand for hydrogen energy to some extent. In 2050, hydrogen consumption (excluding as industrial feedstock) ranges from 0.4-2.3 EJ (PEAK20), 0.4-5.4 EJ (PEAK25) and 0.4-6.0 EJ (PEAK30). The later the peak is reached, the lower the emissions in 2050, and the higher the demand for hydrogen energy. Second, unless the cost of electrolytic hydrogen production falls more than expected (70% lower than today by 2050), hydrogen energy will gain momentum primarily in areas lacking low-cost abatement options. For example, more than 80% of the hydrogen energy is used in the transport sector, about 10% in industry, and the rest in the power and building sectors. For the 25th-75th percentile cases of the PEAK25 scenario, hydrogen fuel cell vehicles account for 33% (over a 10-64% range) of roadway freight transport (Supplementary Fig. 15), and hydrogen direct reduced iron (DRI) technology shares 8% (over a 4-37% range) of the iron & steel production (Supplementary Fig. 16). For PEAK25 scenario, although hydrogen-powered aircraft may emerge after 2035, even under the strictest carbon budget constraints, their share will not exceed 41% in 2050 (Supplementary Fig. 17). Third, for the source of hydrogen, the model gives very consistent results. In China, electrolytic hydrogen production from renewable energy sources (green hydrogen) will become mainstream.

- **Manuscript, Page 24, Line 442-450**

The fourteen selected vital parameters are the technological cost of BECCS (**BECCS Cost**), utilizable biomass resource potential (**Bio Cap**), PV power cost (**Solar Cost**), PV power economic installed capacity (**Solar Cap**), wind power cost (**Wind Cost**), wind power economic installed capacity (**Wind Cap**), energy storage cost (**Storage Cost**), nuclear power cost (**Nuclear Cost**), nuclear power economic installed capacity (**Nuclear Cap**), thermal power with CCS cost (**Thermal CCS Cost**), industry CCS cost (**Industry CCS Cost**), hydrogen production, storage and transportation cost (**Hydrogen Cost**), and price elasticity of energy service demands (**Price Elasticity**).

- **Fig. 6**

Added uncertainty analysis of the hydrogen energy.

Fig. 6 Uncertainty analysis of the hydrogen energy. **a** shows the development of the hydrogen energy (unit: EJ). The box plot shows the first, intermediate range (IQR), and third quantile of all the results, where the data range within 1.5 times the IQR is shown by whiskers. The thick grey line represents the pathway for the intermediate case of each scenario. The divergent colour from blue to green reflects the increasing stringency in the cumulative carbon budget. The cumulative CO₂ parameter corresponds to the absolute value of China's cumulative carbon budget for 2010-2050. **b** presents a breakdown of the different uses of hydrogen energy and the production of hydrogen from electrolysis under the intermediate cases. **c** represents the relationship of the development of the hydrogen energy with input uncertainty parameters and CO₂ emissions. Three parameters with significant impact among fourteen uncertain input parameters are shown for each technology. The remaining results are shown in Supplementary Fig. 12. The variables (except for the CO₂ emissions) represent multiples of the intermediate cases for each scenario. The intermediate case of PEAK20, PEAK25, and PEAK30 are devoted by "□", "+", and "×", respectively.

● **Supplementary Fig. 11**

Added the relationship plot of industry fossil fuel use with CCS and uncertain variables.

Supplementary Fig. 11 Scatter plot and linear regression results between industry fossil fuel use with CCS and Latin hypercube sampling uncertain variables. The intermediate cases of PEAK20, PEAK25, and PEAK30 are represented by "□", "+", and "×".

● **Supplementary Fig. 12**

Added the relationship plot of hydrogen energy use and uncertain variables.

Supplementary Fig. 12 Scatter plot and linear regression results between hydrogen energy use and Latin hypercube sampling uncertain variables. The intermediate cases of PEAK20, PEAK25, and PEAK30 are represented by "□", "+", and "×".

● **Supplementary Fig. 15**

Added fuel mix in the transportation sector.

Supplementary Fig. 15 The fuel mix of road passenger (Panel a) and road freight (Panel b) transport. The box plot shows the first, intermediate range (IQR), and third quantile of all the results, where the data range within 1.5 times the IQR is shown by whiskers. The thick blue line represents the pathway of the NDC scenario, and the thick grey line represents the pathway for the intermediate case of each scenario. The divergent colour from blue to green reflects the increasing stringency in the cumulative carbon budget. The cumulative CO₂ parameter corresponds to the absolute value of China's cumulative carbon budget for 2010-2050.

- **Supplementary Fig. 16**
Added fuel mix in the transportation sector.

Supplementary Fig. 16 Production share of different technologies for iron (Panel a) and cement (Panel b) making. The box plot shows the first, intermediate range (IQR), and third quantile of all the results, where the data range within 1.5 times the IQR is shown by whiskers. The thick blue line represents the pathway of the NDC scenario, and the thick grey line represents the pathway for the intermediate case of each scenario. The divergent colour from blue to green reflects the increasing stringency in the cumulative carbon budget. The cumulative CO₂ parameter corresponds to the absolute value of China's cumulative carbon budget for 2010-2050.

● **Supplementary Fig. 17**
 Added fuel mix in the transportation sector.

Supplementary Fig. 17 The fuel mix of passenger (Panel a) and freight (Panel b) aviation. The box plot shows the first, intermediate range (IQR), and third quantile of all the results, where the data range within 1.5 times the IQR is shown by whiskers. The thick blue line represents the pathway of the NDC scenario, and the thick grey line represents the pathway for the intermediate case of each scenario. The divergent colour from blue to green reflects the increasing stringency in the cumulative carbon budget. The cumulative CO₂ parameter corresponds to the absolute value of China's cumulative carbon budget for 2010-2050.

Comment 3.4:

Forth, I do not see a tight bond between the main analysis part of the paper and the part of discussion and policy insight. The current discussion part is rather general. The authors may want to revise the discussion part to well reflect their key and relevant findings in the analysis part.

Response:

Thank you very much for pointing out our shortcomings in explaining the practical implications of the paper.

We have revised the *Discussion and policy insight* section, summarizing the key findings obtained in the previous study and adding more policy-guidance-rich statements. In our responses to Comment 1.6 and 2.3, you can also find our modifications to the policy insights. Specifically, we have summarized the following five policy insights.

First, we spotlight the positive implications of an enhanced NDC for China to achieve carbon neutrality. From the comparison of PEAK25 and PEAK30 scenarios, we found both China and the international community would benefit greatly from China reaching its carbon peak early and at a lower level of emissions.

Second, the significant impact of cumulative emissions on the emission reduction pathway can be seen in the paper. Therefore, it is very important to consider total carbon emission control target after 2030 rather than carbon emission intensity reduction target alone. And our paper also illustrates the importance of establishing a cumulative emission target to anchor China's emissions pathway.

Third, identifying the important role of renewable energy, nuclear energy, energy storage, hydrogen and CCS and the significant benefits of cost reductions, we propose to develop a diverse technology portfolio and promote technological innovation to stockpile future technologies.

Fourth, understanding that decarbonization in China is not only limited to the energy supply sector, but that demand management and lifestyle changes in the demand sectors also play a large role, we emphasize sector-wide continuous efforts. While the energy supply sector needs to do more in the near term, emissions reductions in the energy demand sector should not be ignored, and cross-sectoral cooperation can be more effective in sharing the cost of reductions.

Fifth, climate governance and energy transition are closely related to investment and industry development, and we call for a moratorium on new thermal power plants, and thermal power plants under construction should be capture-ready to reduce the financial risk. Over the next 10 years, the annual investment in renewable power will be at least twice the amount invested in all power sources in 2019. We therefore recommend vigorously developing green finance and reducing fossil energy investments. Meanwhile, international climate investment and financing support will be strengthened to help achieve the ambitious goals.

Changes/additions to the manuscript:

- **Manuscript, Page 18-19, Line 331-335**

Although the future of policy and technology development are full of uncertainties, some important consensus can be obtained from this study. Most importantly, decisive action and clear goals enable us to realize carbon neutrality. In particular, clean energy supply, low-carbon energy consumption, and green energy investment go hand in hand on the road to carbon neutrality.

- **Manuscript, Page 19, Line 336-346**

Net-zero emission targets should not distract from the urgent need for deep emission reductions. Understanding that energy system transition cannot be accomplished overnight, decisive actions facilitate an earlier emission decline, which buys time for China's transition to a low-carbon economy at a domestic level. If China's CO₂ peaks in 2025 instead of 2030, welfare losses will be reduced by at least 50%, marginal abatement costs in 2050 will be kept within reasonable limits and heavy reliance on carbon removal technologies will be avoided. At an international level, these actions lay a foundation for China to make a more ambitious contribution to mitigating global climate change. With a peak in 2030, it is almost impossible for China to achieve cumulative emissions below 250 Gt in 2010-2050, meaning that either the 1.5-degree target called for in many studies will be difficult to achieve, or China will have to generate significant negative emissions for a long time in the second half of this century.

- **Manuscript, Page 19-20, Line 354-361**

Policymakers need to consider setting total emission control target and cumulative emission target after 2030 rather than carbon emission intensity reduction target alone to encourage progressive emission reductions. Net-zero emission targets alone, while cumulative emissions remain highly uncertain, will ultimately lead to large differences in emission reduction pathways, technology choice and transition costs. For China, an appropriate cumulative emissions target should dovetail with the announced carbon neutrality target in a way that does not impose an excessive burden on society, but also appropriately bears international responsibility and contributes to the 1.5-degree target.

- **Manuscript, Page 20, Line 362-371**

Key technologies to achieve carbon neutrality have not been fully developed, and it is important to cultivate diverse technologies and build a portfolio of carbon neutral technology reserves. Innovation in renewables and advanced technologies is a critical backbone of the energy transition. The rapid reduction in PV and wind power costs has already been changing the landscape for addressing climate change. Considering the massive demand for variable renewable energy,

further technological progress in wind, solar, and energy storage will powerfully impact the transition cost. Hydrogen, biomass and CCS technologies, which in the past have often been used as back-up resources, will be important pillars of carbon neutrality due to their indispensable role in a net-zero and carbon-negative world, hence the urgent need for sustained R&D investments and industrial pilots for these high-cost technologies.

- **Manuscript, Page 21, Line 380-385**

Investment is the vane of future development. The results have clearly indicated that existing thermal power has large risk of capital stranding. Hence, we urge a moratorium on new thermal power, and thermal power plants under construction should be capture-ready. In the context of carbon neutrality, it is imperative to reduce fossil energy investments, support renewable energy expansion, and promote international cooperation.

Peer review further comments –

Reviewer #1 (Remarks to the Author):

The authors have addressed most of my comments.

I still have a few questions on the model data and assumptions which might involve significant remodeling but none of those impact the main insights of the paper. I, therefore, do not think it is necessary to do so.

One small suggestion to the manuscript: try to avoid using words "undoubtedly", "obvious" when presenting the results or policy recommendation, and do not overstate the findings. As the authors also noted, this is a "what-if" study, it highlights the impact of uncertainties.

Reviewer #2 (Remarks to the Author):

My previous concerns have been well answered and addressed by the authors and I do not have any further comments.

Reviewer #3 (Remarks to the Author):

The authors have clearly expended considerable effort in revising their manuscript, and they have done an excellent job. The manuscript is much improved as a result.

Response to Reviewers

NCOMMS-21-14011A

November 22, 2021

Reviewer #1

Comment:

The authors have addressed most of my comments. I still have a few questions on the model data and assumptions which might involve significant remodeling but none of those impact the main insights of the paper. I, therefore, do not think it is necessary to do so. One small suggestion to the manuscript: try to avoid using words "undoubtedly", "obvious" when presenting the results or policy recommendation, and do not overstate the findings. As the authors also noted, this is a "what-if" study, it highlights the impact of uncertainties.

Response:

Thank you very much for your comments, which have led to a significant improvement of our paper. We agree with you about avoiding absolute and subjective wording, and we have carefully revised the paper. In addition, we sent the manuscript to the editor's recommended language editing service (Nature Research Editing Service) to present our work more clearly.

Reviewer #2:

Comment:

My previous concerns have been well answered and addressed by the authors and I do not have any further comments.

Response:

We are very grateful for your valuable comments, which help us a lot to improve the quality of our papers.

Reviewer #3

Comment:

The authors have clearly expended considerable effort in revising their manuscript, and they have done an excellent job. The manuscript is much improved as a result.

Response:

Thank you for your recognition of our research work and your comments have been of great help in enriching the content of our paper.